# Tunable and functional deep eutectic solvents for lignocellulose valorization

Yongzhuang Liu[1,2], Noemi Deak [3], Zhiwen Wang [4], Haipeng Yu[1], Lisanne Hameleers[5], Edita Jurak [5], Peter J. Deuss [4] & Katalin Barta [2,3✉]

Stabilization of reactive intermediates is an enabling concept in biomass fractionation and depolymerization. Deep eutectic solvents (DES) are intriguing green reaction media for biomass processing; however undesired lignin condensation is a typical drawback for most acid-based DES fractionation processes. Here we describe ternary DES systems composed of choline chloride and oxalic acid, additionally incorporating ethylene glycol (or other diols) that provide the desired 'stabilization' function for efficient lignocellulose fractionation, preserving the quality of all lignocellulose constituents. The obtained ethylene-glycol protected lignin displays high β-O-4 content (up to 53 per 100 aromatic units) and can be readily depolymerized to distinct monophenolic products. The cellulose residues, free from condensed lignin particles, deliver up to 95.9 ± 2.12% glucose yield upon enzymatic digestion. The DES can be recovered with high yield and purity and re-used with good efficiency. Notably, we have shown that the reactivity of the β-O-4 linkage in model compounds can be steered towards either cleavage or stabilization, depending on DES composition, demonstrating the advantage of the modular DES composition.

[1] Key Laboratory of Bio-Based Material Science and Technology, Ministry of Education, Northeast Forestry University, Harbin, P. R. China. [2] Stratingh Institute for Chemistry, University of Groningen, Groningen, The Netherlands. [3] Karl-Franzens University of Graz, Institute of Chemistry, Graz, Austria. [4] Department of Chemical Engineering (ENTEG), University of Groningen, Groningen, The Netherlands. [5] Department of Bioproduct Engineering (ENTEG), University of Groningen, Groningen, The Netherlands. ✉email: katalin.barta@uni-graz.at

Task-specific, benign reaction media, suitable for the fractionation and depolymerization of renewable starting materials can have a significant impact on the sustainability and industrial viability of future biorefineries. This development requires the integrated use of green solvents for efficient biomass processing[1–4].

Stabilization[5] of reactive intermediates during lignocellulose fractionation and lignin depolymerization is a powerful strategy to prevent undesired recondensation phenomena in order to increase the yield of desired platform chemicals, especially lignin-derived aromatics[4–9].

Due to their favorable physical and chemical properties, e.g. negligible vapor pressure, recyclability, and non-toxic character, deep eutectic solvents (DES) have emerged as alternatives for conventional organic solvents[10]. DES can be easily prepared from a wide variety of hydrogen bond donors (HBD) and hydrogen bond acceptors (HBA)[11,12]. Owing to this modularity, DES are excellent candidates for the incorporation of additional functions, for example, the 'stabilization function', highly desired in biorefining[5]. An elegant example of task-specific DES is described by Singh and coworkers for closed-loop biorefinery strategies[13,14]. However, true 'designer' DES systems that comprise catalytic and stabilization functions to obtain non-condensed lignin structures suitable for selective depolymerization; or have the ability to control product outcomes in depolymerization/fractionation reactions, have not yet been developed.

Recognizing the excellent solvent strength and ability to break the strong interactions between lignin and cellulose, many binary and ternary DES compositions were applied for lignocellulose fractionation with promising results[15–22]. These systems may comprise choline chloride ChCl as HBD and organic acids or alcohols as HBA or incorporate mineral or Lewis acid additives. A detailed discussion on compositions and uses of relevant systems is included in Supplementary Note 1. While these systems generally deliver good quality cellulose, the lignins obtained this way typically suffer undesired structural modification (affected by the acid content of the DES) via significant loss of their valuable aryl-ether structure and the formation of recalcitrant C–C bonds, which renders them poorly suitable for further catalytic depolymerization (Fig. 1a)[16,18,23].

We have previously described that in situ trapping the reactive C2-aldehydes originating from acidolysis of β-O-4 linkages by various diols leads to cyclic C2-acetals and suppression of recondensation processes (Fig. 1b) during lignin depolymerization[7,24]. These approaches require the use of strong acids such as trifluoromethanesulfonic acid[7] or metal triflates as catalyst[25–29], stoichiometric amounts of diols as reagents (in case of C2-cyclic acetal products), and toluene or 1,4-dioxane as solvents. Our aim here was to replace these corrosive strong acids and conventional organic solvents and integrate the acidolysis and stabilization functions directly into the DES composition to result in recyclable, non-toxic, and bio-derived alternative reaction media, 'tailor made' for lignin depolymerization and lignocellulose fractionation.

Here, we present the design of inexpensive and potentially bio-based (approximately $600–800/ton, Supplementary Note 2, Supplementary Table 1) ternary deep eutectic solvent systems consisting of ChCl as HBA as well as oxalic acid (OA) and a varying amount of diols as HBDs, which allow for tuning the reactivity of lignin β-O-4 model compounds (Fig. 1c) either to bond cleavage/stabilization or selective derivatization/protection by careful selection of the type and relative ratio of DES components. The same ternary DES composition is known for the extraction of flavonoids and alkaloids from plants[30,31].

Next, based on the selective derivatization/protection concept, a lignocellulose fractionation protocol was established (100 °C for 24 h), which lead to the isolation of high-quality cellulose and protected, uncondensed lignin that delivered high aromatic monomer yields upon catalytic depolymerization. The EG-protected lignins may find application in the field of material and polymer science[32,33] and their increased polarity leads to the development of a fractionation protocol that uses less water and allows for facile DES recycling.

## Results

**Development of tunable DES systems for lignin stabilization.** Acid-catalyzed lignin depolymerization generally results in low aromatic monomer yield due to extensive recondensation that occurs under these conditions (Fig. 2a)[34–39]. We have previously shown that unstable **5a** undergoes recondensation reactions that can be suppressed by trapping in the form of cyclic acetals by a variety of diols (e.g. ethylene glycol)[7]. Another way that alcohols and diols can stabilize lignin is by incorporation via capturing the formed carbocation species under acidic conditions (Fig. 2b). This also partly prevents the lignin β-O-4 from breakdown, the extent of which depends on reaction conditions[5,27,28,33,40–42].

Based on this reaction network we first set out to establish an acidic DES system that allows for direct involvement in the above-mentioned stabilization. Initially, the cleavage of the β-O-4 linkage in a model compound **1a** without the need for strong acid (triflic acid or triflates) was investigated. (For all details related to model compound studies please see Supplementary Note 3). Inspired by the report on lignocellulose pulping with choline chloride/oxalic acid (ChCl/OA) DES systems[18], in which the linkages of the lignocellulose carbohydrate complex (LCC) as well as most of the lignin β-O-4 linkages were efficiently cleaved, we have started our investigation by treating **1a** with ChCl/OA (1:1 molar ratio) DES at 100 °C, for 2 h. Indeed, full **1a** conversion was observed, while the selectivity to guaiacol and aldehyde (**5a**) were 84% and 20%, respectively, showing rapid condensation of the aldehyde **5a** as expected (Supplementary Figs. 5 and 6). The role of the acid in the scission of **1a** was confirmed by observing no conversion in choline chloride/ethylene glycol (ChCl/EG) DES as shown in Supplementary Fig. 7.

*Cleavage/stabilization pathway in (CS-DES).* After establishing successful cleavage of **1a** in the (ChCl/OA) DES, we investigated the ternary composition ChCl/OA/EG (1:1:2 molar ratio), named cleavage/stabilization DES (CS-DES), essentially incorporating the stabilization function directly into this DES by adding an appropriate amount of ethylene glycol (for analysis of ternary DES see Supplementary Note 4). Gratifyingly, at 100 °C for 20 min, 56% conversion of **1a** was observed, together with 21% selectivity of acetal **6aa**, showing that trapping of the C2-aldehyde is in principle successful (Supplementary Fig. 8). Additionally, the ethylene glycol incorporated **4aa** was the major product at this temperature showing also that the reactive benzylic carbocation is trapped by this DES providing additional suppression of condensation pathways. To see if the outcome could be further steered to the cleavage products the temperature was increased to 120 and 140 °C showing a satisfactory full conversion of **1a**, 71% yield of guaiacol, and 67% yield of acetal **6aa** (Fig. 3b and Supplementary Table 2). A gradual increase of the acetal was observed with increasing the amount of EG (**2a**) at 120 °C with the best yield of 67% was observed with 60 mmol EG (ChCl/OA/EG, 1:1:2 molar ratio) in the CS-DES system (Fig. 3c and Supplementary Table 3). Time-course experiments revealed that 20 min is necessary to achieve full conversion at 120 °C (Fig. 3d and Supplementary Table 4). From these experiments, it can be concluded that the highest selectivity of both guaiacol and acetal **6aa** were 71% and 67% with full conversion **1a** at 120 °C for 20 min. These results surpass those we have previously achieved in

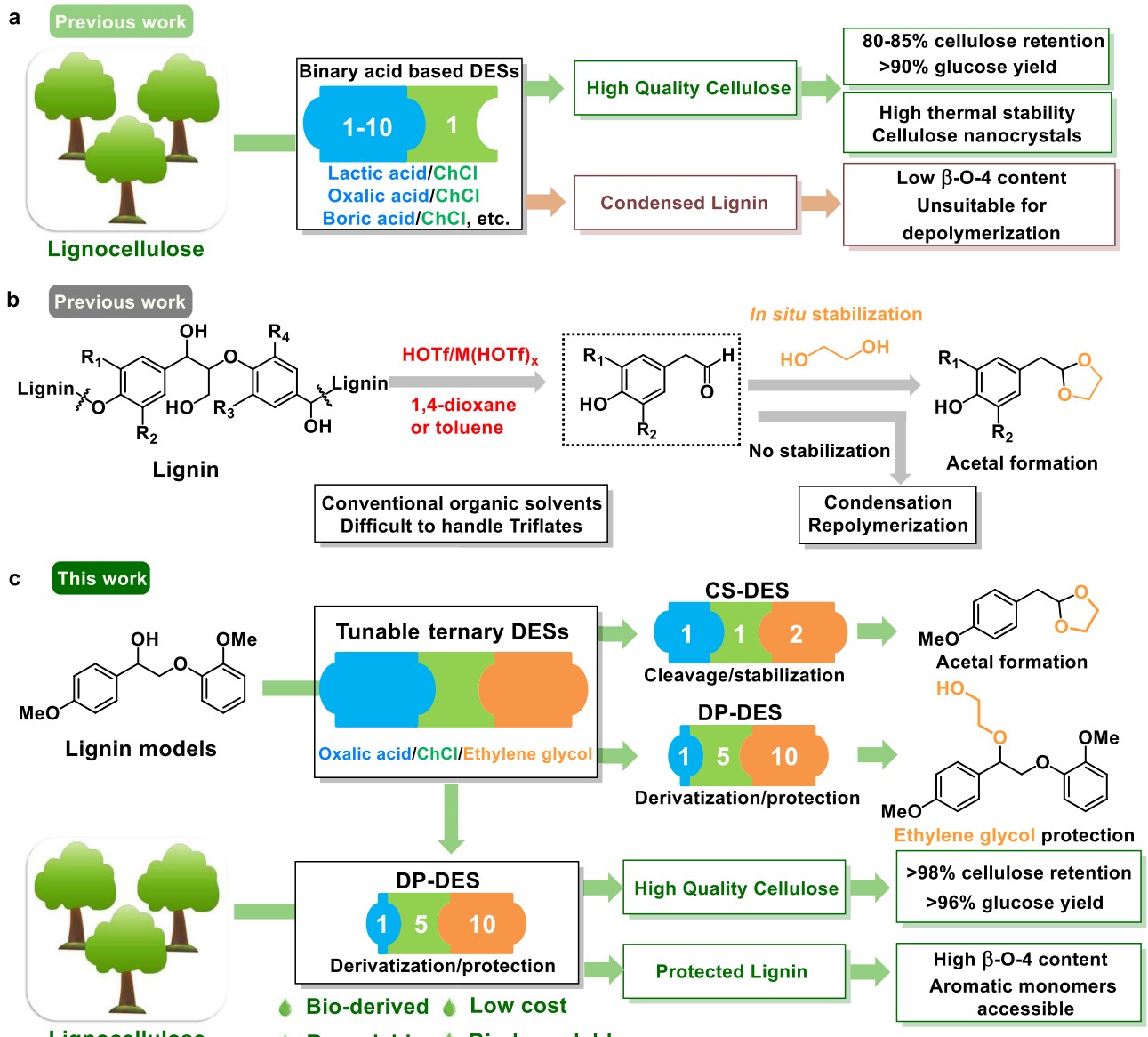

**Fig. 1 Development of tunable and functional DESs for lignocellulose valorization. a** Binary acid-based deep eutectic solvents (ChCl as HBA and oxalic acid or lactic acid as HBDs) for lignocellulose fractionation resulting in good quality cellulose (80–85% cellulose retention), which can be digested to >90% glucose or nanocellulose and condensed lignin fragments with low β-O-4[16,18]. **b** Capturing reactive intermediates in the acid-catalyzed scission of β-O-4 linkages during lignin depolymerization in order to prevent recondensation phenomena. Instable C2-aldehydes are trapped in the form of their cyclic C2-acetals using diols. Reactions were carried out in conventional organic solvents using strong acids. **c** This work: Tunable ternary DES systems (choline chloride, ChCl as HBA and oxalic acid, OA and ethylene glycol, EG as HBDs) for controlled lignin depolymerization and lignocellulose fractionation as environmentally benign alternatives for conventional organic solvents. Control over the product outcome (cleavage/stabilization versus derivatization/protection) in acidolysis of lignin model compounds by adjusting the relative ratios of HBA and HBD. Efficient lignocellulose fractionation with integrated DES recycling and isolation of high-quality cellulose (>98% cellulose retention), which can be digested to >96% glucose as well as protected, high β-O-4 content lignin highly suited for depolymerization to aromatic monomers.

functional IL systems with ethylene glycol incorporation and are lower than those obtained in conventional organic solvents[43]. We attribute the product yields below 100% to the existence of side reactions or alternative trapping pathways due to the highly ionic nature of the DES solvent media.

The data also confirm that species **4aa** is formed as the initial product prior to the β-O-4 linkage scission (Supplementary Fig. 8). Generally, using this particular set of DES compositions, the selectivity toward the ethylene glycol incorporated species **4aa** was low, however, the fact that it could be detected lead us to investigate the tunability of the DES and reaction conditions, in

order to suppress the cleavage reaction and favor ethylene glycol incorporation at the benzylic carbocation.

*Derivatization/protection pathway in (DP-DES).* In order to render the derivatization/protection pathway (Fig. 3a) dominant and prevent possible β-O-4 cleavage, the acidity of the reaction media was tuned by decreasing the amount of OA in DESs (33 wt % in CS-DES versus 1 wt% in DP-DES) at milder reaction temperatures 80 and 100 °C (Fig. 3e and Supplementary Table 5), using substrate **1a**. An excellent, 95% selectivity of **4aa** was obtained at 80 °C and the desired ethylene glycol incorporated

**Fig. 2 Reaction pathways in lignin acidolysis. a** Recondensation pathways in acidolysis demonstrated on a model compound comprising the β-O-4 moiety. These involve rapid dehydration and direct capture of the formed carbocation species, by a neighboring aromatic lignin moiety or by intramolecular condensation to result in recalcitrant C–C linkages[37-39]. Alternatively, depolymerization occurs via a series of steps involving the vinyl ether intermediate **3a** which suffers rehydration and cleavage to furnish a free phenol and the corresponding C2 aldehyde **5a** (see also Supplementary Fig. 4). **b** Incorporation of alcohols or diols into lignin during the acid-catalyzed organosolv pulping process.

product could be obtained with a 70% isolated yield (Supplementary Figs. 9 and 10, Supplementary Note 3.1.4), a decrease in **4aa** selectivity and simultaneous increase in the cleavage products (guaiacol and **6aa**) were observed at 100 °C. The fraction of OA in the DP-DES was varied from 1 to 10 wt% and reactions were carried out at 80 °C for 6 h (Fig. 3f). These experiments revealed that **4aa** product yield is only mildly affected by the OA amount as long as the reaction temperature is kept relatively low, signifying ineffective cleavage of **1a** and **4aa** at temperatures below 100 °C (Fig. 3f and Supplementary Table 6). Time-course experiments carried out in 1 wt% OA containing DP-DES at 80 °C (Fig. 3g and Supplementary Table 7) were in accordance with these findings whereby limited decomposition of the incorporation product **4aa** was observed even after 24 h reaction time.

**Reaction scope with respect to different diols using DP-DES.** The DP-DES system was extended to using a variety of lignin model compounds as well as a range of diols. First, model compound **1a** was treated with previously optimized DP-DES composition (ChCl, 30 mmol: diol, 60 mmol: OA, 0.6 mmol) based on glycerol (**2b**), 1,3-propanediol (**2c**), as well as 1,2-butanediol (**2d**). The corresponding diol-incorporated products (**4ab**, **4ac**, and **4ad**) could be obtained in good to moderate isolated yield as shown in Fig. 4. Interestingly, with glycerol incorporation, a total 73% isolated product yield was obtained as a mixture of two regioisomers (see Supplementary Figs. 11 and 12, Supplementary Note 3.1.6) **4ab** and **4ab′** (97% and 3%, respectively, by GC-FID). The yields of **4ac** (41%, see Supplementary Fig. 13) and **4ad** (18%, see Supplementary Fig. 14) were lower due to the decreased substrate conversion, which could be attributed to the increased hydrophobicity of the used diols (**2c**, **2d**) that might affect viscosity and ion mobility as suggested by the Hole Theory[44].

The diol incorporation strategy was also successful with a variety of model compounds that closely represent the native lignin β-O-4 motif and are known to be more labile. Using **1b** and **1c** as model compounds and ethylene glycol containing DP-DES (ChCl, 4.2 g, 30 mmol: OA, 0.6 mmol, 0.08 g, 1 wt%: EG, 60 mmol, 3.6 g) at 80 °C for 24 h, the desired products **4ba** (Supplementary Note 3.1.5) and **4ca** have obtained in 76% and 71% yields, respectively. Starting with model compound **1d** containing a free phenol moiety, **4da** was obtained in an isolated yield of 82% (Supplementary Note 3.1.7). All ethylene-glycol-protected compounds were additionally characterized by heteronuclear single quantum coherence spectroscopy (HSQC) NMR and GC/MS (Supplementary Figs. 15–18).

Thus overall, the ChCl/OA/EG ternary DES system can be tuned to give selective diol and polyol incorporation at the benzylic position by a catalytic amount of OA at mild conditions, on the other hand increasing the amount of OA and/or the temperature leads to the breakdown of the β-O-4 motif.

During/after cleavage (cyclic acetal formation) and stabilization (benzyl ether formation), a portion of the EG present in the parent ternary DES actually gets incorporated into the corresponding lignin-derived products by the formation of covalent bonds. Detailed NMR and spectroscopic experiments (Supplementary Note 4.1, Tables 8 and 9, Figs. 19–23), as well as calculations (Supplementary Note 4.2, Table 10, Figs. 24–27), confirmed the ternary nature of the parent DES while revealing a stronger interaction between ChCl and OA than between ChCl and EG.

**Fractionation for high-quality lignin and DES recycling.** Lignocellulose fractionation has been extensively studied in binary DES, mainly ChCl/lactic acid[15,16] and ChCl/OA DES[18] (see also Fig. 1a) as well as ChCl/boric acid DES[19]. The lignin–carbohydrate complex (LCC) in lignocellulose was efficiently cleaved resulting in high-quality cellulose and condensed

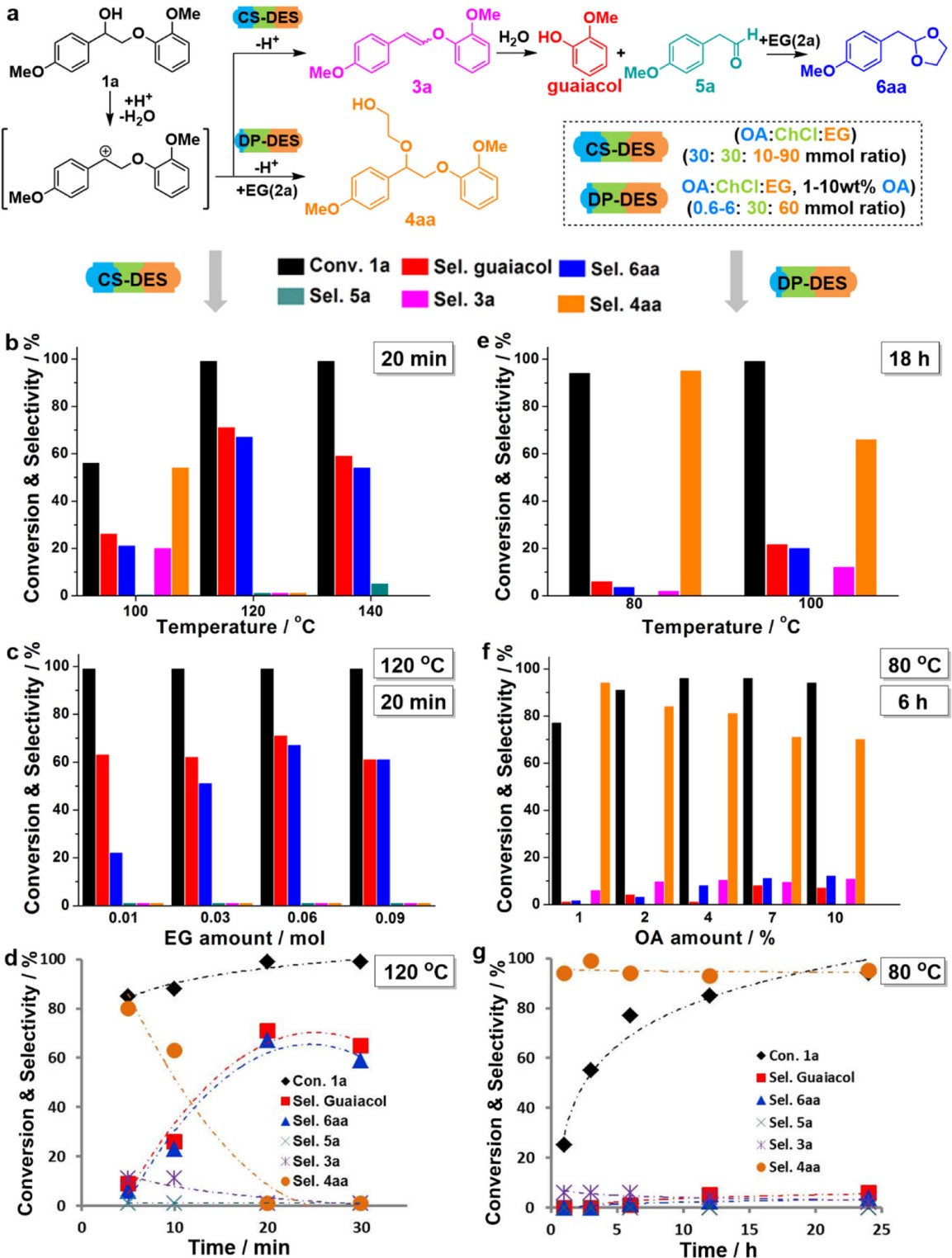

**Fig. 3 Development of different DES systems by model compound study. a** Reaction pathways during treatment of β-O-4 lignin model compound **1a** in CS-DES or DP-DES: the sequence starts with rapid dehydration of **1a**. In CS-DES: intermediate **3a** is formed; rehydration and cleavage lead to the formation of guaiacol and **5a**, unstable **5a** is trapped in the form of cyclic acetal **6aa** by the ethylene glycol present in CS-DES. In DP-DES: cleavage is suppressed by a controlled amount of OA, **4aa** is formed by the attack of ethylene glycol on the carbocation intermediate originating from **1a**. Optimization of **1a** (25 mg) reaction in CS-DES (ChCl, 4.2 g, 30 mmol: OA, 30 mmol, 3.8 g: EG, 60 mmol, 3.6 g) by **b**. The temperature of 100–140 °C for 20 min; **c** EG amount (10–90 mmol) in CS-DES at 120 °C for 20 min; **d** Reaction time from 5 to 30 min at 120 °C. Optimization of **1a** (50 mg) reaction in DP-DES (ChCl, 4.2 g, 30 mmol: OA, 0.6 mmol, 0.08 g, 1 wt%: EG, 60 mol, 3.6 g). **e** Temperature of 80–100 °C for 18 h; **f** OA amount (0.6–60 mmol, 0.08–0.8 g, 1–10 wt%) in DP-DES at 80 °C for 6 h; **g** Reaction time of 1–24 h at 80 °C. Conversion of **1a** and selectivity of the products were calculated by internal standard calibration with octadecane based on GC-FID analysis; products were identified by GC/MS; DCM and water were used to extract the products from DESs.

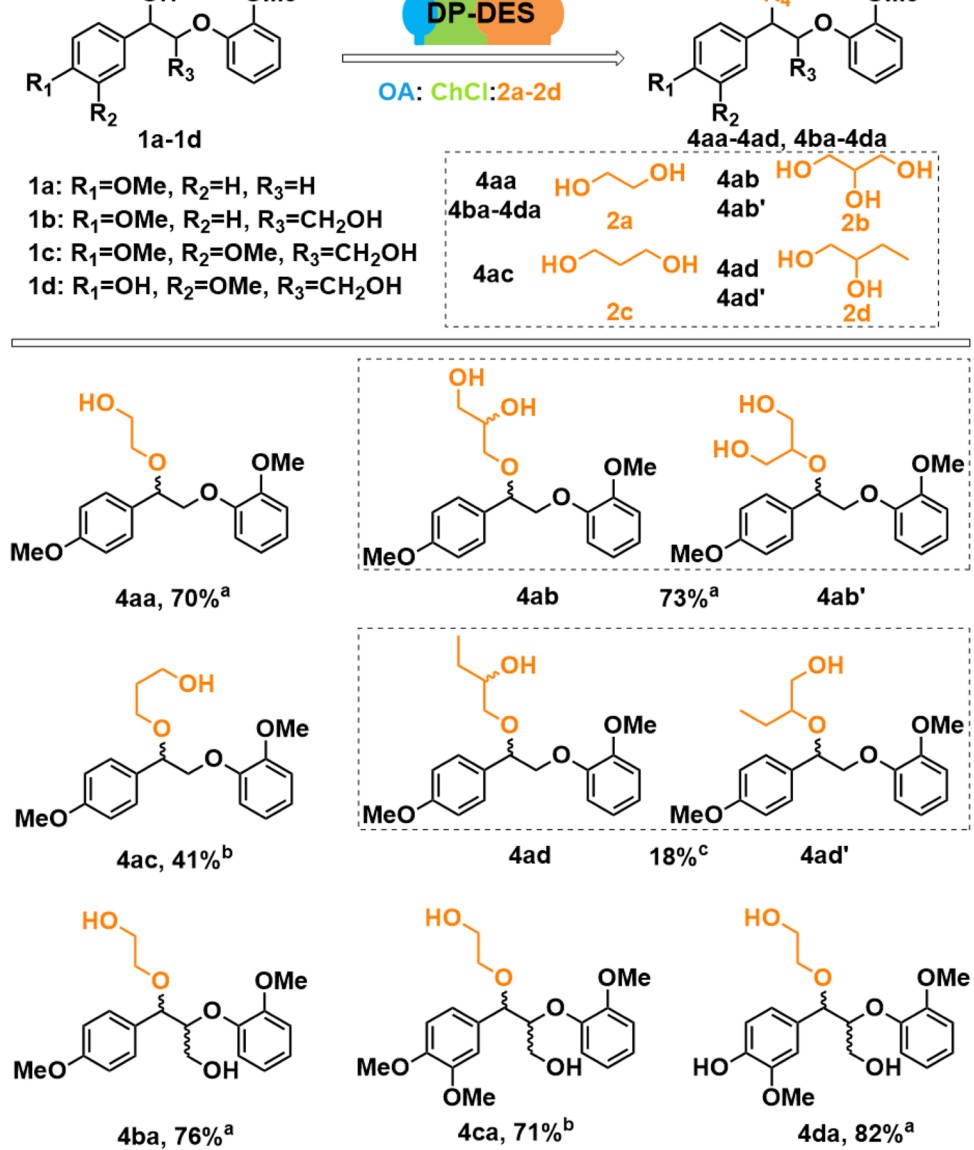

**Fig. 4 Reaction scope of diol protection protocol using ternary DP-DES systems.** The incorporation of various diols (glycerol, **2b**; 1,3 propanediol, **2c**; 1,2 butanediol, **2d**) into β-O-4 lignin model compound **1a** or incorporation of ethylene glycol (**2a**) into various lignin model compounds (**1b**, **1c**, and **1d**) were achieved in DP-DES (30 mmol ChCl: 60 mmol diols (**2a–2d**): 1 wt% OA), 80 °C, 24 h, 50 mg various lignin model compounds (**1a–1d**) was used. [a]Signifies isolated yield, ratio of **4ab** and **4ab′** is 97:3 determined by GC-FID; [b]Product yield determined by GC-FID, product identification by GC/MS; [c]Mixture of regioisomers, not isolated, ratio not quantified.

organosolv lignin in which most of the β-O-4 linkages were cleaved. With the tunable DES systems established above using model compounds, our aim was to maintain the efficiency of lignocellulose fractionation and at the same time, preserve the value of both the cellulose as well as the lignin component by suppressing recondensation phenomena. We have first selected the derivatization/deprotection DES (DP-DES) with EG (ChCl, 120 mmol, 16.8 g: EG, 240 mmol, 14.4 g: 1 wt% OA, 2.5 mmol, 0.32 g) described above, in order to study potential EG protection of lignin during lignocellulose fractionation. First, 2 g birch lignocellulose was subjected to treatment in ~32 g DP-DES at 80 °C for 24 h. Previous DES fractionation studies followed a product isolation protocol that includes cellulose separation by organic solvent/water as antisolvents followed by precipitation of lignin in water (Fig. 5a). Following this protocol to obtain the DP-DES fractionated cellulose

using ethanol as anti-solvent and lignin by water precipitation method, only a small amount of lignin precipitated even when the OA amount in DP-DES increased from 1 to 10 wt% as shown in Supplementary Fig. 28. Considering the complexity of lignocellulose, the DES acidity was increased by increasing the OA content in the DP-DES to 10 wt% (DP-DES10). Gratifyingly, when fractionation was carried out at 100 °C for 24 h the amount of recovered precipitated lignin significantly increased (Supplementary Fig. 28). However, only 15 wt% (yield based on Klason lignin of the birch starting material) of "black" condensed lignin was obtained after filtration, meanwhile, centrifugation resulted in a stable dispersion of lignin in water (Supplementary Fig. 29). Considering that our goal with the use of the DP-DES systems was to protect the lignin structure from acidolysis and recondensation by incorporating diols into its structure, increased hydrophilicity and water

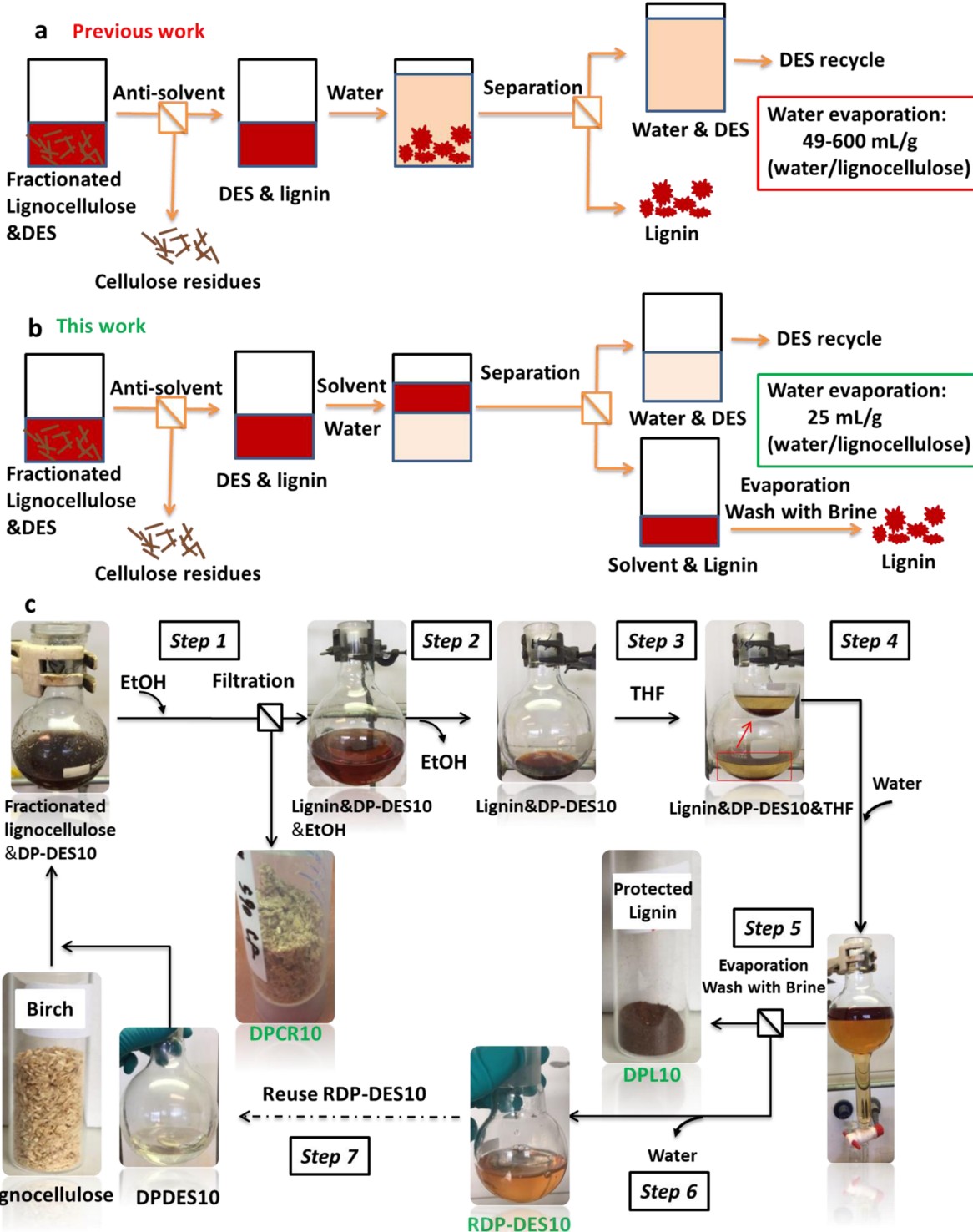

**Fig. 5 Lignocellulose fractionation and lignin isolation protocol in DP-DES. a** Traditional lignin isolation method from DES. Typically, after the separation of cellulose, lignin is precipitated from DES by the addition of water followed by filtration or centrifugation/decantation, and the DES is recovered by water removal. **b** The lignin isolation method is based on liquid/liquid extraction from DES. After adding organic solvent and a small amount of water, lignin is selectively extracted from a concentrated ionic aqueous DES solution and is subsequently recovered after evaporating the organic solvent. High purity DES is recycled by water removal. **c** Systematic birch lignocellulose fractionation protocol in DP-DES10 (ChCl, 120 mmol, 16.8 g: EG, 240 mmol, 14.4 g: 10 wt% OA, 25 mmol, 3.2 g) by steps. Carbohydrate-rich residues are isolated using ethanol as an anti-solvent after filtration (Step 1). Ethanol is evaporated to obtain a lignin/DES mixture (Step 2). Upon addition of THF as well as water, a brown-colored THF phase containing lignin is clearly separated from the aqueous phase containing DES. The lignin is isolated by extraction of an aqueous DES phase with THF (Steps 3 and 4). DPL10 was obtained as powder after solvent removal and precipitated in brine (Step 5). DP-DES10 is recycled by removal of water by distillation (Step 6). The recycled DP-DES10 was characterized by yield (mass percentage to original DP-DES10) and purity (Supplementary Note 6.1.3) and reused for a second run (Step 7).

solubility of this fractionated lignin, compared to those obtained by conventional DES treatment, was expected.

Assuming that the poor isolated yield of lignin was caused by retention of a significant amount of the 'protected lignin' in the aqueous phase, an alternative fractionation protocol for liquid/liquid extraction of the diol-incorporated lignin Fig. 5b was developed. After evaluating a number of solvents (Supplementary Figs. 30–32), THF/water co-solvent was chosen for the isolation of the diol-incorporated lignin.

With this method in hand, separation of cellulose and lignin, as well as efficient recycling and reuse of the applied DP-DES was established (Fig. 5c, Steps 1–7). Gratifyingly, using DP-DES10 (ChCl, 120 mmol, 16.8 g: EG, 240 mmol, 14.4 g: 10 wt% OA, 25 mmol, 3.2 g), this method delivered a significantly higher yield of protected lignin (DPL10) after Step 5 (Table 1, Entry 2, 40.0 ± 3.5 wt% by THF/water extraction versus 15 wt% by water precipitation), meanwhile, 67.8 ± 0.80 wt% of cellulose residue (DPCR10) also remained after Step 2. A unique advantage of this protocol is the use of significantly less water (~25 mL water per gram lignocellulose) compared to the classical isolation methods (~49–600 mL water per gram lignocellulose)[16,18,20,23]. Another key point of the method is the ease of recycling of the applied DES. After fractionation, the DP-DES10 could be recovered in a high yield (93.4%) after water removal (Step 6). The performance of the recycled RDP-DES10 in a new run of lignocellulose fractionation gave nearly identical results, yielding 41 wt% of lignin (RDPL10, see Table 1, Entry 3). A second round of recycling also resulted in

a similarly high, 93%, DES recovery. The RDP-DES10 recovered after two fractionation runs were of high purity as evidenced by [1]H NMR spectroscopy (Supplementary Note 6).

Naturally, during EG protection, a portion of the EG present in the DES is incorporated into the lignin structure, therefore the change of DES composition upon fractionation and recycling was carefully assessed (Supplementary Note 6, Figs. 33–36 and Tables 11 and 12). Overall, the loss of ChCl was found minimal, while the loss of EG and OA was more significant, but quantifiable after the first recycling, and can be compensated by simply adjusting the composition of RDP-DES10 by adding appropriate amounts of these components. Moreover, the purity of the recycled DES remains reasonably high and the RDP-DES10 (even with lower EG and OA content) can be reused with good efficiency (Supplementary Fig. 37).

Further, the lignin yield was increased to 53.3 ± 10.7 wt% (Table 1, Entry 4, DPL20) by tuning the composition of DP-DES through increasing the acid content to 20 wt% OA (DP-DES20). For comparison, the classical DES ChCl/OA[18], as well as the above-described cleavage/stabilization DES (CS-DES), were evaluated. Using the ChCl/OA DES system ChCl/OAL lignin was obtained in 51 wt% yields, which is expected (based on literature) to be low in aryl-ether content and have an overall condensed character (Table 1, Entry 5). Using CS-DES at 120 °C, CSL120 was obtained in 44% yield (Table 1, Entry 6) and fractionation in CS-DES at 80 °C for 24 h resulted in comparable 43 wt% lignin (CSL80) yield (Table 1, Entry 7). Notably, after fractionation in CS- and DP-DES, the obtained lignins displayed

---

**Table 1 Yield and semi-quantitative analysis of lignins obtained by various DES treatments of birch lignocellulose.**

| | DP-DES | ChCl/OA DES | CS-DES |
|---|---|---|---|
| | OA:ChCl:EG | OA:ChCl | OA:ChCl:EG |
| | Entry 2, 3 and 4 | Entry 5 | Entry 6 and 7 |

| Entry | DES system | T / °C | Lignin yield / %[a] | Delignification / %[b] | Content of linkages per $Au_{100}$[c] β-O-4 | β'-O-4 | Condensation / %[c] | Lignin abbreviation |
|---|---|---|---|---|---|---|---|---|
| 1 | - | - | - | - | 63 | - | 1 | MWL[d] |
| 2 | DP-DES10 | 100 | 40.0±3.5[g] | 56.0±2.14[i] | 7-8 | 40-45 | 14-16 | DPL10 |
| 3 | RDP-DES10[e] | 100 | 41 | 56 | 21 | 30 | 15 | RDPL10 |
| 4 | DP-DES20 | 100 | 53.3±10.7[g,h] | 67.9±4.62[i] | 4-6[j] | 26-38[j] | 27-40 | DPL20 |
| 5 | ChCl/OA DES[f] | 120 | 51 | 49.6±3.68[i] | 4 | - | 53 | ChCl/OAL |
| 6 | CS-DES[f] | 120 | 44 | 61.4±1.92[i] | 9 | 23 | 23 | CSL120 |
| 7 | CS-DES | 80 | 43 | 68.5±3.55[i] | 12 | 35 | 17 | CSL80 |

*General conditions*: 2 g birch lignocellulose was used for fractionation in ~32 g **DP-DES** at 80–100 °C for 24 h. After fractionation, the obtained lignin was isolated by THF/water extraction method.
[a]The yield of lignin was calculated based on the Klason lignin content in birch after correcting for the alcohol incorporation[45].
[b]Removed lignin based on the Klason lignin content of the solid residue.
[c]Determined by semi-quantitative HSQC NMR integration using cross signals at the α position in the side-chain region to the integration of total 100 aromatic units ($Au_{100}$) in the aromatic region. Total aromatic = (($S_{2/6}$ + $S'_{2/6}$)/2 + $S_{condensed}$) + ($G_2$ + $G_5$ + $G_6$)/3), β-O-4 content = (β-O-4α)/total aromatic × 100, β'-O-4 content = (β'-O-4α)/total aromatic × 100. Condensation: $S_{condensed}$% = $S_{condensed}$/total $S_{units}$ × 100.
[d]Milled wood lignin (MWL) was used as the control, the MWL was typically isolated from milled birch powder by dioxane/water extraction as previously reported[46].
[e]Average from a set of 4 extractions showing standard deviations.
[f]Average from 2–7 extractions showing standard deviations.
[g]Recycled DP-DES10 was used for a second fractionation process.
[h]Yield likely varies due to work-up and different amounts of impurities.
[i]High variation between different batches.
[j]Fractionation was conducted with a reaction time of 30 min.
*Note*: DP-DES(X) represented that derivatization/protection DES was prepared with ChCl/EG/OA (120 mmol ChCl, 240 mmol EG, (X) wt% OA); CS-DES represented that cleavage/stabilization DES was prepared with ChCl/EG/OA (120 mmol ChCl, 240 mmol EG, 120 mmol OA); ChCl/OA DES was prepared with ChCl and OA (120 mmol ChCl and 120 mmol OA).

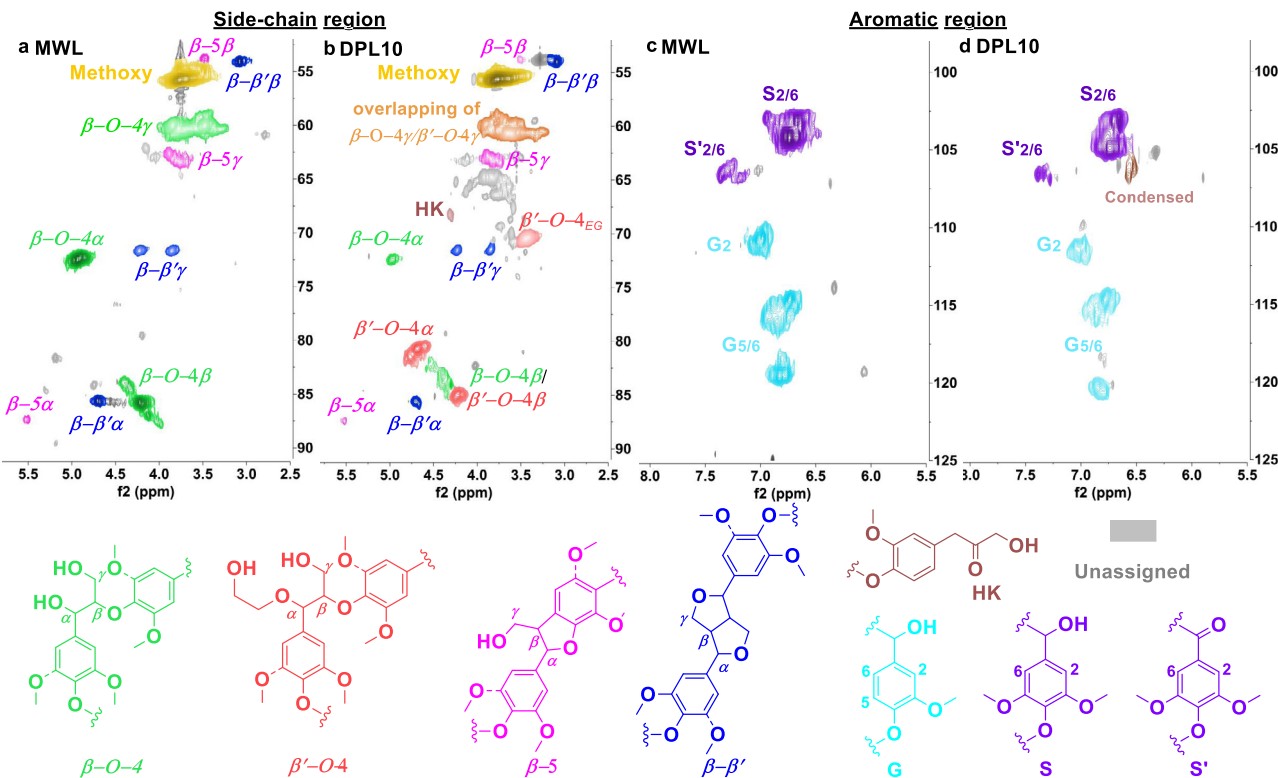

**Fig. 6 Comparison of mill wood lignin (MWL) and DPL10 from birch lignocellulose.** $^{1}$H–$^{13}$C cross signals in the side-chain region of MWL (**a**) and DPL10 (**b**), showing obvious ethylene glycol incorporation signals of β′-O-4 which means stabilization of the benzylic carbocation in β-O-4 by EG (the EG incorporation signals were confirmed by overlapping the HSQC NMR spectrum of lignin and model compound) and aromatic region of MWL (**c**) and DPL10 (**d**) with typical S and G signals, slight condensation (14–16%) signal was observed in the aromatic region of DPL10. Spectra of both MWL and DPL10 were recorded by dissolving the corresponding lignin in d6-acetone for measurement instead of commonly used d6-dmso (several drops of $D_2O$ were added when not fully soluble in d6-acetone)[68].

significantly lighter color compared to the ChCl/OA DES, which is associated with less condensation (see Supplementary Fig. 38). We attribute this to the beneficial stabilization effect of DP-DES preventing C–C bond formation reactions by either trapping of the benzylic carbocation and/or the formed reactive aldehyde fragments via acetal formation as discussed in more detail above[7,24].

**Structural characterization and comparison of DES-lignins.** The lignins obtained upon treatment in various DES systems shown in Table 1 were structurally characterized in terms of linkage distribution, extent of condensation as well as ethylene glycol incorporation. The observation of negligible nitrogen content by elemental analysis confirmed that there is no choline incorporation in DPL10 (Supplementary Methods 1.2). Ethylene glycol incorporation into DPL10 was confirmed by comparing the HSQC NMR spectra with that of **4ba** (see Supplementary Fig. 39), showing clearly overlapping $^{13}$C–$^{1}$H cross signals belonging to the respective α, β, γ— moieties as well as for the incorporated ethylene glycol moiety. As a control, ChCl/OAL isolated by traditional water precipitation, represented condensed lignin (Supplementary Fig. 40), clearly displaying signals at $^{1}$H (6.35–6.65 ppm) and $^{13}$C (106–109 ppm) in the aromatic region that are assigned to condensed S units (53% $S_{condensed}$ compared to uncondensed S units) and weak signals in the region corresponding to the β-O-4 moiety. These data are fully consistent with previous reports on structural characterization of condensed lignin obtained by ChCl/OA or ChCl/Lactic acid DES system[15,18]. In order to assess the level of ethylene glycol incorporation, the HSQC NMR spectrum of DPL10, (Entry 2) was compared to a representative HSQC NMR spectrum of birch milled wood lignin (MWL, Entry 1, see Supplementary Methods 1.3 for

detailed description and Supplementary Table 13 for signal assignment). In the HSQC NMR spectrum of MWL (Fig. 6a), signals typical for the β-O-4, β-β, and β-5 linkages were observed with a β-O-4 content of 63 per 100 aromatic units ($Au_{100}$). In the side-chain region of DPL10 (Fig. 6b) obtained upon DP-DES10 fractionation, signals belonging to ethylene glycol incorporation are clearly observed (β′-O-4 signal, Fig. 6b) with a high content of 40–45 per $Au_{100}$ with a combined β-O-4 and β′-O-4 content of 47–53 per $Au_{100}$ (Table 1, Entry 2). The aromatic region of both MWL and DPL10 show signals typical for S and G units (Fig. 6c, d), albeit weak condensation signals (14–16% $S_{condensed}$) in the spectrum of DPL10 were also observed. Furthermore, lignin RDPL10 obtained upon fractionation in the recycled DP-DES10 showed an increased β-O-4 content of 21 per $Au_{100}$ while a decrease β′-O-4 of 30 per $Au_{100}$ (Table 1, Entry 3, Supplementary Fig. 41). The decreased β′-O-4 can be attributed to lower acidity of the RDP-DES10 due to the consumption of OA in DP-DES10 during the first run. When further increasing the OA amount to 20 wt% in the DP-DES, good lignin yield (53.3 ± 10.7 wt% of DPL20) but more condensation (27–40% $S_{condensed}$) were detected (Supplementary Fig. 42). Using CS-DES, previously established for the cleavage of model compounds followed by protection of the products, for lignocellulose fractionation at 120 °C for 30 min or at 80 °C for 24 h resulted in lignins still displaying a good β′-O-4 content of 23 for (CSL120) and 35 $Au_{100}$ for (CSL80), respectively. However, higher temperature (120 °C) fractionations in CS-DES lead to the recovery of lignin with significant condensation (23% $S_{condensed}$) after 30 min (see Table 1, Entries 6 and 7, Supplementary Figs. 43 and 44). Overall, it can be concluded that a higher amount of OA in DES or higher reaction temperature led to lignin condensation. Ethylene

glycol incorporation was observed both in low OA DP-DES as well as high OA CS-DES fractionated lignin. Protected lignins DPL10 and CSL80 fractionated with less OA or at milder temperature displayed a high β′-O-4 content and less condensation, potentially attractive for application in selective depolymerization to specific aromatic monomers (see below).

Furthermore, the higher extent of condensation in ChCl/OAL compared to DPL10 was additionally shown by assessing their thermal stability by thermogravimetric analysis (Supplementary Figs. 45 and 46) and pyrolysis GC–MS (Supplementary Fig. 47 and Table 14).

**Catalytic depolymerization of DES-lignins.** To show the potential of the developed protection methodology for lignin depolymerization, the DES-treated lignins were subjected to catalytic hydrogenolysis and acidolysis protocols established previously[25,26,47]. The yield and distribution of the desired aromatic monomers are shown in Tables 2 and 3, respectively[6,7]. Hydrogenolysis was carried out using Ru/C (5 wt% Ru loading), 40 bar $H_2$, methanol as solvent at 200 or 220 °C for 18 h. While condensed lignin (ChCl/OAL) delivered only a poor total monomer yield of 4 wt% in accordance with the extensive condensation of 53% and relatively low amount of β-O-4 linkages in its structure, 'EG-protected' DPL10 resulted in a high combined yield (22–24 wt%) of aromatic monomers (Table 2, Entries 2 and 3). Similarly, the CS-DES-fractionated lignin (CSL80 for example, 17% condensation) that was also characterized by a high β-O-4 content resulted in a good, 20 wt% monomer yield. In comparison, the total monomer yield from CSL80 or DPL10 was roughly 5 or 6 times higher than that from ChCl/OAL. While this yield is somewhat lower than previously reported Ru/C catalyzed hydrogenolysis of birch lignin isolated by enzymatic hydrolysis and mild acidolysis extraction (~32 wt% monomers yield, Ru/C catalyst, 30 bar $H_2$, 220 °C), our method delivers comparable aromatic product yield as well as high-quality cellulose instead of dilute glucose solution and at much shorter processing time (24 h instead of several days)[48]. Higher monomer yields can be only achieved by Ru/C or Pd/C catalyzed "lignin first" birch lignocellulose hydrogenolysis (40–52 C% aromatic monomers) under similar conditions[47,49].

Next, EG-protected (DPL10) lignin was subjected to the acidolysis/stabilization protocol previously described by our group[25,26]. After optimization, 13.8 wt% yields of two specific acetal monomers (8AG and 8AS) were obtained using 10 wt% Fe(OTf)₃, 60 wt% EG at 140 °C for 15 min. as shown in detail in Table 3. These C2 acetals were shown to be the major products from the total monomer distribution. While the presence of C3 ketal (8AS′) as a minor product was derived from the related Hibbert's ketone. The monomer ratio of S/G was 4.8, which was a bit lower than that in DPL10 (S/G: 5.5 calculated by 2D HSQC NMR). Overall, the optimized monomer yields obtained from DPL10 are similar to those obtained from mild organosolv extraction (but obtained at much lower yield) and significantly higher than those from condensed lignins, consistently with our previous findings[26,41].

In conclusion, the incorporation of ethylene glycol moieties into lignins during lignocellulose fractionation was successfully carried out by using carefully adjusted DP-DES or CS-DES compositions. These lignin structures feature a high β′-O-4 content, resembling a stabilized native β-O-4 structure and an overall low fraction of C–C bond linkages, which is indispensable for efficient depolymerization. Overall, the relatively high yield and excellent β-O-4 content of the lignins obtained upon fractionation with the DP-DES systems, and the good yield and excellent selectivity of the aromatic monomer products achieved after depolymerization show promise of the developed stabilization method for lignin valorization. Optimization may focus on increasing the yield of DP-DES10 isolated lignin that shows the highest β-O-4 content. A comparison with other catalytic systems can be found in Supplementary Table 15. Moreover, these lignins possess

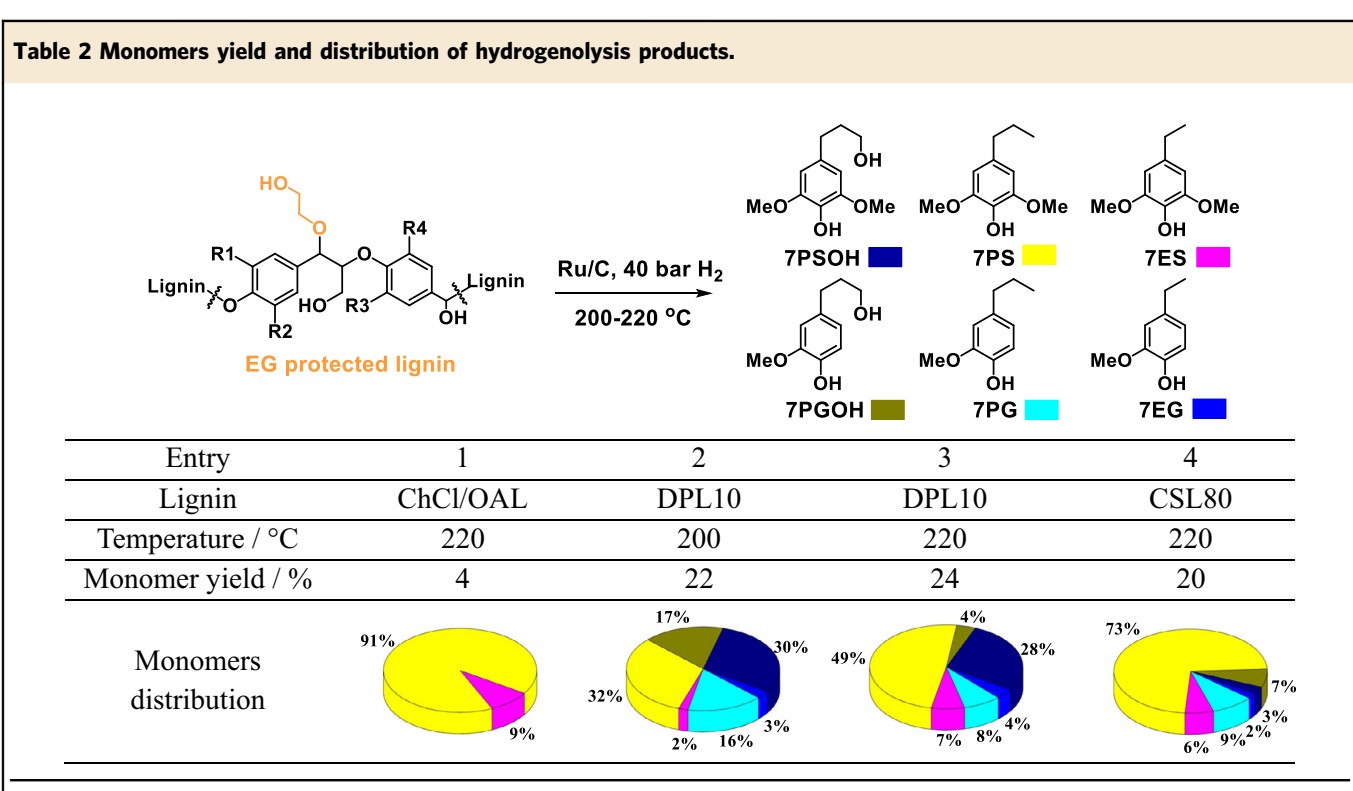

**Table 2 Monomers yield and distribution of hydrogenolysis products.**

| Entry | 1 | 2 | 3 | 4 |
|---|---|---|---|---|
| Lignin | ChCl/OAL | DPL10 | DPL10 | CSL80 |
| Temperature / °C | 220 | 200 | 220 | 220 |
| Monomer yield / % | 4 | 22 | 24 | 20 |

*General condition*: 200 mg DES fractionated lignin, 200 mg Ru/C (5 wt% Ru loading), 20 mL methanol, 18 h, 20 mg 3,5-dimethylphenol was used as internal standard, products were identified by GC/MS and monomers yield were calculated based on GC-FID calibration, using calibration with authentic standards.

**Table 3 Monomers yield and distribution of acidolysis products.**

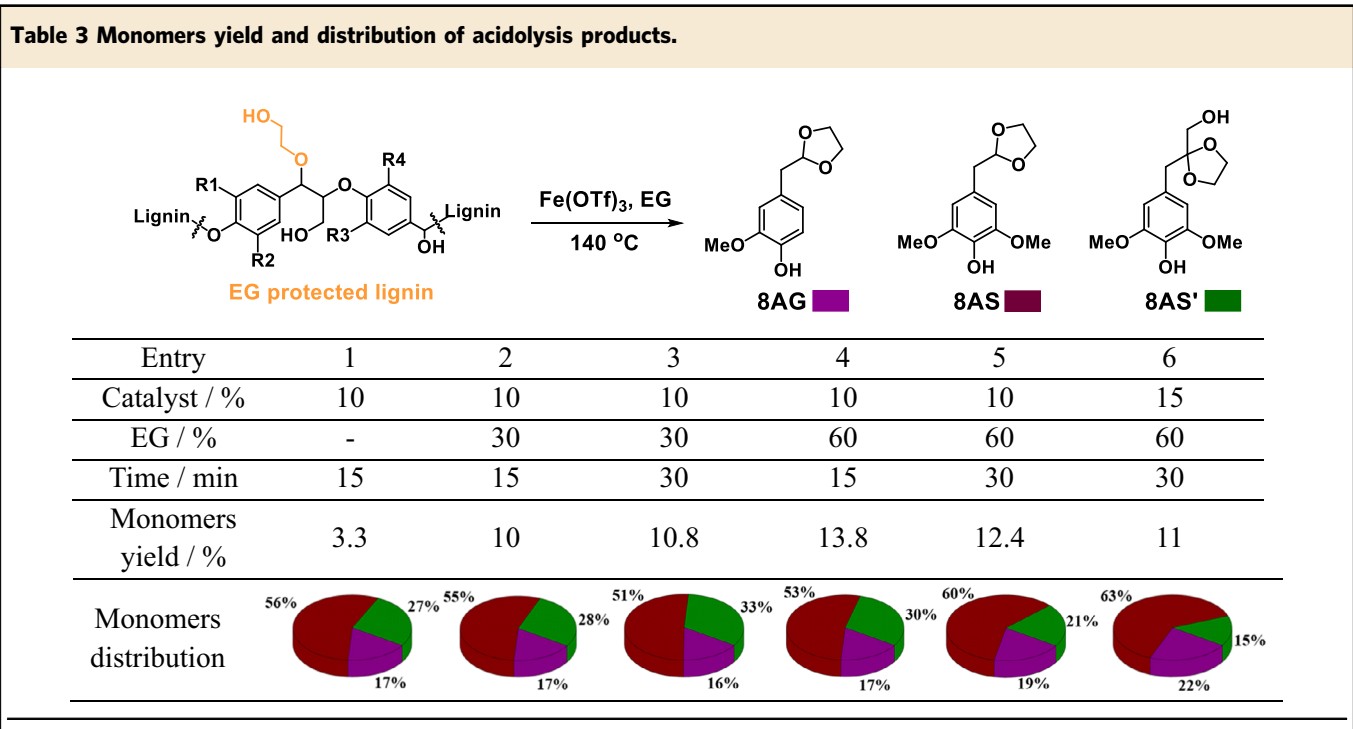

| Entry | 1 | 2 | 3 | 4 | 5 | 6 |
|---|---|---|---|---|---|---|
| Catalyst / % | 10 | 10 | 10 | 10 | 10 | 15 |
| EG / % | - | 30 | 30 | 60 | 60 | 60 |
| Time / min | 15 | 15 | 30 | 15 | 30 | 30 |
| Monomers yield / % | 3.3 | 10 | 10.8 | 13.8 | 12.4 | 11 |

General condition: 50 mg DPL10, 0.9 mL dioxane, 20 μL octadecane (0.25 M), and 30 or 60 wt% EG (based on lignin) were mixed in a 20 mL glass vial with a septum cap, the vial was put into a pre-heated aluminum heating block at 140 °C, 100 μL Fe(OTf)3 (50 mg/mL) was added by syringe, reacted for 15 or 30 min. Products were identified using GC-MS and quantified using GC-FID using calibration with authentic standards.

interesting properties such as increased hydrophilicity and more –OH functionalities and can thus serve as valuable starting materials for the synthesis of polymers and materials[32,33] or other specialized purposes.

**Characterization and valorization of the cellulose residues.** Next, the quality of cellulose residues (CR) obtained upon fractionation was investigated by compositional analysis (Supplementary Fig. 48 and Table 16). The use of all the ternary DES (DP or CS-DES) led to excellent cellulose retention (up to 98%), accompanied by a relatively high (up to 44%) hemicellulose retention depending on DES composition (Supplementary Table 17). We attribute the latter observation to the nature of the fractionation procedure used.

The behavior of cellulose and hemicellulose was separately investigated by treating MCC and xylan as models in DP-DES10 and DP-DES20 (Supplementary Note 5 and Fig. 49) showing near complete MCC retention (99% and 94%) as well as full xylan dissolution with no accompanying coloring, and partial xylose hydrolysis (7%).

In-depth sugar composition analysis using HPAEC-PAD of the raw birch lignocellulose and the cellulose residues obtained upon the various DES treatments (Supplementary Note 5.3.2 and Table 18) showed a loss in total hemicellulose content, while the ternary DES treatment retained more hemicellulose (~38% w/w) compared to the binary DES treatment (23% w/w). It was revealed that hemicellulose suffers debranching upon treatment with all DES systems (loss of Gal, Ara, Rha, GlcA and partial depolymerization to xylose), but remains most intact with DP-DES10. This is in line with high hemicellulose retention. It is also this DES system that preserves the β-O-4 content of lignin and which leads to the highest retention of sugars.

The delignification increased with more OA in the ternary DES systems, with the highest delignification of 67.9 ± 4.62% achieved in DP-DES20. The CRs were promising starting materials for saccharification.

The enzymatic hydrolysis activity of the CRs obtained upon fractionation in different DES compositions was investigated and compared to untreated birch lignocellulose (see Supplementary Methods 1.4 for details). The results are summarized in Fig. 7a, b, and Supplementary Tables 19 and 20. CR obtained from treatment with binary ChCl/OA DES delivered 58.8 ± 5.01% glucose yield and 75.3 ± 1.40% xylose yield upon saccharification for 96 h. However, DPCR10, showed excellent enzymatic hydrolysis activity, resulting in superb 95.9 ± 2.12% glucose and 86.7 ± 1.53% xylose yield. Nevertheless, DPCR20 (obtained using DP-DES20 containing more OA) showed a slight decrease in sugars yield: 92.5 ± 1.63% for glucose and 83.9 ± 1.44% for xylose, respectively. Furthermore, CRs obtained by fractionation using CS-DES systems at 80 °C for 30 min (CSCR80) or 120 °C for 30 min (CSCR120), were characterized by medium performance (87.8 ± 2.77% glucose, 84.9 ± 0.94% xylose and 80.1 ± 1.77% glucose, 78.6 ± 1.89% xylose yield, respectively).

It has to be mentioned that previously, comparable glucose yield was reported from ChCl-OA DES pretreated corncob (45.2% compared to 58.8 ± 5.01% in this work)[50], albeit at different fractionation conditions. Furthermore, the extent of delignification in the obtained CR processed in different DES compositions seems to have no direct correlation with the glucose yields obtained [delignification DPCR10 = 56.0 ± 2.14%; DPCR20 = 67.9 ± 4.62% at 95.9 ± 2.12% and 92.5 ± 1.63% glucose yield, respectively]. Indeed, several papers report high glucose yield (~90%) with 50–60% delignification[17,51,52], even 91% glucose yield at ~10% delignification[53]. Therefore, we believe that the formation of condensed lignin particles, in analogy to the previously reported inhibiting effect of redeposited pseudolignin[54–56], is the reason for decreased enzyme activity. Thus CR obtained in DP-DES10 and DP-DES20 with significant stabilization effect gave the best glucose yields.

In order to gain more insight into the morphology of the obtained CR, structural characterization by scanning electron microscopy (SEM) was performed (Fig. 7). Untreated birch

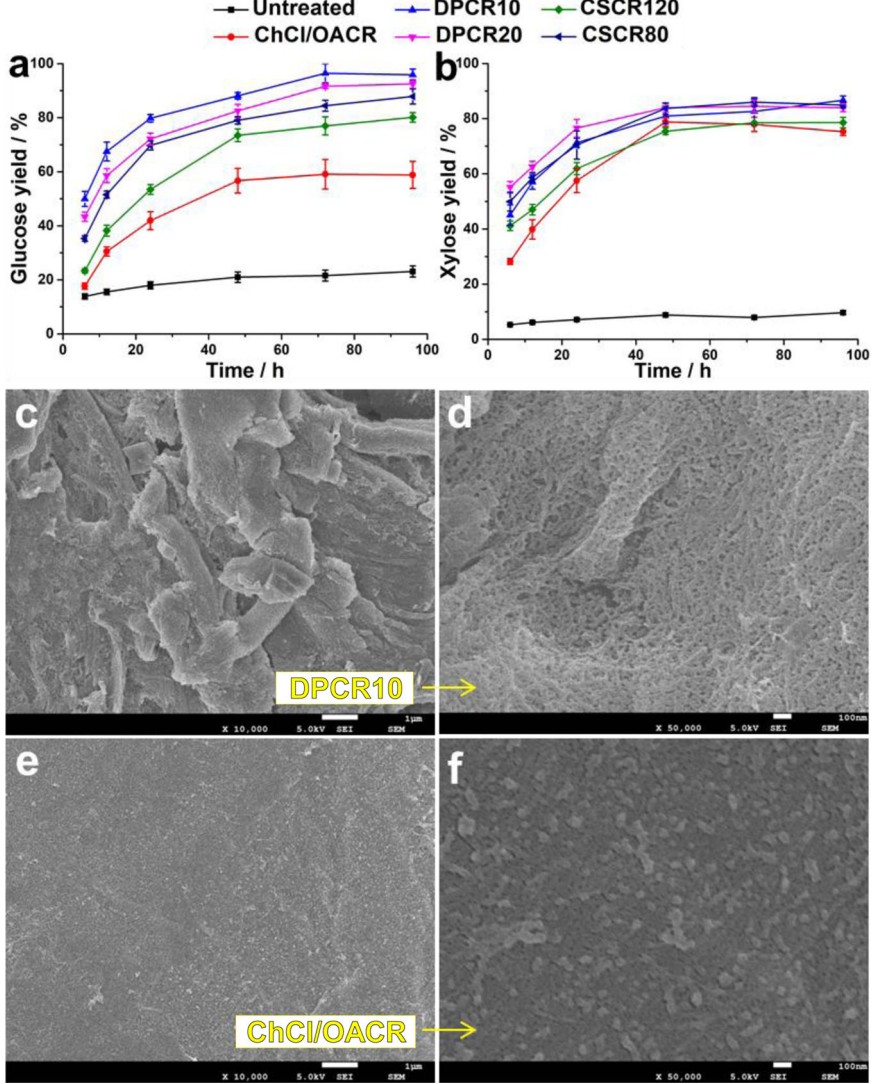

**Fig. 7 Enzymolysis and SEM images of DES fractionated cellulose residues.** Enzymatic hydrolysis yield of glucose (**a**) and xylose (**b**) from untreated birch lignocellulose and various DES fractionated CRs. The yield of glucose and xylose was calculated based on the cellulose and hemicellulose content in various substrates. The yields were average values from a set of at least three experiments showing standard deviations. SEM images of DP-DES10 fractionated CR at low magnification (**c**) and high magnification (**d**), the cell structure of DPCR10 was fibrillated and without formation of lignin particles; ChCl/OA DES fractionated CR at low magnification (**e**) and high magnification (**f**), showing obvious lignin particles formation (condensed lignin which is confirmed as shown in Supplementary Section 5.2.1, the average diameter of ~54 nm) on the surface.

lignocellulose showed a typical wood cell structure and smooth surface as shown in Supplementary Fig. 50. When ternary DES (DP-DES10) was used, the basic wood cell structure of DPCR10 appeared to be disrupted at low magnification, which can indeed aid the action of the enzymes (Fig. 7c). At high magnification, the DPCR10 showed a smooth surface and intervascular pitting without any particle formation. Interestingly, the DPCR10 was partially fractionated to nanofibers as shown in Fig. 7d. In contrast, when the conventional binary ChCl/OA DES was used for fractionation, as a control reaction, the ChCl/OACR remained an obvious wood cell structure at low magnification (Fig. 7e), however, multiple nanosized particles were observed throughout the surface of ChCl/OACR at high magnification (Fig. 7f). Earlier pioneering work has visualized pseudo-lignin or lignin droplets by SEM, albeit under different processing conditions[55,57]. In order to clarify the origin of these particles in our system, microcrystalline cellulose (MCC) was treated under fractionation conditions alone or together with xylan or organosolv lignin as

model compounds in DP-DES versus ChCl/OA DES (Supplementary Note 5.2.1) and the CR residues were imaged by SEM (Supplementary Figs. 51 and 52). Based on these experiments, we attribute the formation of these nanosized particles to condensed lignin rather than pseudo-lignin[58].

Overall, the exceptional enzymatic hydrolysis activity of DPCR10 was credited to the ability of the used ternary DP-DES10 to restrain the condensed lignin formation in the CRs[59].

## Discussion
By developing ternary DES compositions that involve diols beside oxalic acid/choline chloride, here we achieve the incorporation of a stabilization function, essential in biorefining, in DES systems. Notably, we were able to control the reactivity of the β-O-4 moiety to result in either in scission or protection in model compounds by simply tuning the relative ratios of the DES components (Fig. 3). For lignocellulose, the system showed

excellent fractionation efficiency under mild conditions, while maintaining the value of both the cellulose as well as lignin constituents. In the lignins so obtained, the native β-O-4 moiety was largely preserved due to EG stabilization, while diol incorporation was tunable by variation of DES composition. These lignins delivered 6 times more aromatic monomers upon depolymerization via hydrogenolysis than the condensed analogs obtained using binary (ChCl/OA) DES and good yields of aromatic monomers from acidolysis. These monomers could be further converted to a number of attractive products, e.g. as previously shown toward phenol[60] or to higher-value molecules such as pharmaceuticals or biologically active molecules[61–63], enhancing the economic feasibility of this approach[64]. This is further enhanced by the excellent glucose yield obtained from the cellulose residues. When comparing the quality of the cellulose residues both by SEM analysis and enzymatic hydrolysis, a clear difference was observed: The ternary DES was able to completely suppress condensed lignin formation and result in high glucose yields (95.9 ± 2.12%) compared to (58.8 ± 5.01%) for the classical (ChCl/OA) DES.

Concerns about adverse environmental impact, high cost of solvents used for biomass fractionation, and energy requirements/ material loss during their recycling are key bottlenecks toward achieving economically feasible and sustainable biorefineries[65]. The DES developed in this work holds the potential to overcome these limitations. Multiple studies point to low toxicity and good biodegradability of similar classical ChCl/OA DES[66,67]. This is a promising indication for our ternary systems, largely based on the same components. The developed fractionation method uses less water than typical procedures and makes it possible to recycle the DES without any noticeable change in performance, and with good purity and the DES composition is cost-competitive and can be, in the future derived from renewable resources (Supplementary Note 2.2).

In summary, we have developed the first proof of principle multifunctional DES systems that allow for stabilization of reactive intermediates during lignocellulose and lignin processing. While the EG component is responsible for stabilization, the OA component renders the use of any corrosive mineral acids obsolete.

Due to their ease of tunability, and beneficial physicochemical properties, the DES developed herein may hold the potential for their future industrial application in biorefining.

## Methods

**Preparation of DESs.** DES was prepared by mixing choline chloride (ChCl), oxalic acid dihydrate (OA), and ethylene glycol (EG) in a 100 mL round bottom flask, the mixture was heated at 80 °C with stirring until a clear liquid form. Typically for cleavage/stabilization DES (CS-DES), the molar ratio of ChCl, OA, and EG was 30 mmol: 30 mmol: (5–60 mmol); for derivatization/protection DES (DP-DES), the molar ratio of ChCl, OA and EG was 30 mmol: (1–10 wt%, 0.6–6 mmol): 60 mmol. When other diols were used to prepare DP-DES, the same molar ratio of diols was used instead of EG.

**Lignin model compound depolymerization in CS-DESs or DP-DESs.** In a 20 mL microwave glass vial lignin model compound **1a** (25 mg) was mixed with CS-DES (~11.6 g) and the vial was sealed and heated with stirring to the appropriate temperature (80–120 °C) for the indicated reaction time (5–30 min). (*Note:* when the temperature was over 100 °C, a needle was used to release the pressure.) After cooling down in an ice bath, 0.1 mL stock solution of the internal standard (octadecane, ~10 mg/mL) was added and the mixture was stirred for 5 min. Then DCM (20 mL×3) and water (20 mL) were used to extract the products and the combined DCM phase was concentrated under reduced pressure. The product was diluted in 2 mL DCM, filtered through a 0.45 μm PTFE filter and analyzed by GC–MS or GC-FID whereby conversion and selectivity values were calculated based on internal standard and calibration curve.

For the derivatization/protection DES a slightly different protocol was followed. **1a** (50 mg) was mixed with DP-DES (~8 g), the vial was sealed and heated with stirring to the appropriate temperature (80–100 °C) for 24 h. Product extraction and analysis followed the same procedure as in the case of CS-DES.

**Fractionation of birch lignocellulose in DP-DES.** Lignocellulose fractionation in DES was carried out in a 100 mL round bottom flask. Typically, 2 g birch lignocellulose was mixed with DP-DES (~32 g). The flask was stirred and heated to 100 °C for 24 h. After that, the mixture was cooled down and diluted into 50 mL EtOH, the carbohydrate residues (DPCR) were recovered after filtration (washed with extra 50 mL×2 EtOH) and dried in the desiccator. Ethanol was evaporated from the filtrates by rotary evaporation to obtain a mixture of DES and lignin. Subsequently, the mixture was extracted with THF (100 mL×3) and water (50 mL) and the combined THF phase was concentrated by rotary evaporation and washed with Brine to result in lignin (DPL), 180 mg, 40% yield.

**Catalytic hydrogenolysis of DES fractionated lignins.** The catalytic depolymerization of DES fractionated lignin was carried out in a 100 mL high-pressure Parr autoclave equipped with a mechanical stirrer. Typically, the autoclave was charged with Ru/C catalyst (200 mg, 5 wt% Ru loading), DES lignin (200 mg), and methanol (20 mL). The autoclave was sealed, purged with $H_2$ and pressurized with 40 bar $H_2$ at room temperature. The autoclave was heated to 200 or 220 °C and stirred at 400 rpm for 18 h. After the reaction, the reactor was cooled to room temperature, 20 mg 3,5-dimethylphenol as internal standard was added into the solution and stirred for 10 min, then the mixture was centrifuged and filtered through a 0.45 μm PTFE filter, and a 1 mL sample was analyzed by GC–MS and GC-FID.

**Catalytic acidolysis of DES-fractionated lignins.** Typically, 50 mg DPL10, 0.9 mL 1,4-dioxane, 20 μL of 0.25 M *n*-octadecane stock solution in 1,4-dioxane and 30 μL EG (60 wt%) was mixed in a 20 mL glass vial with a stirring bar and sealed with a septum cap. The vial was put into a pre-heated heating block at 140 °C, followed by adding 50 μL of Fe(OTf)$_3$ (100 mg/mL) and reacted for 15 min. After the reaction, the vial was rapidly cooled on ice and the mixture was filtered over a plug of Celite and washed with extra 2 × 2 mL dioxane. The filtrate was concentrated under a high vacuum to get a sticky solid. The sticky solid was suspended in dichloromethane with vortex and sonication after which toluene was added and again mixed properly by vortexing and sonication. The mixture was centrifuged for 2 min at 6000 rpm to separate the clear liquid phase. After repeating the extraction process three times, the combined clear liquid was concentrated under a high vacuum. The products were diluted in 2 mL DCM and filtered through a 0.45 μm PTFE filter before injecting into GC–MS and GC-FID for analysis.

## Data availability

The authors declare that all of the data that support the findings of this study are available within the article and its Supplementary Information files or from the corresponding author upon reasonable request.

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

## Acknowledgements

K.B. is grateful for financial support from the European Research Council, ERC Starting Grant 2015 (CatASus) 638076, and ERC Proof of Concept Grant (PURE) 875649. This work is part of the research program Talent Scheme (Vidi) with project number 723.015.005 (K.B.), which is partly financed by the Netherlands Organization for Scientific Research (NWO). H.Y. and Y.L. are grateful for the support by the National Natural Science Foundation of China (Grant nos. 31925028, 32001280). Y.L. and Z.W. are grateful for financial support from the China Scholarship Council (Grant nos. Z.W.: 201706300138, Y.L.: 201706600008).

## Author contributions

Y.L. designed the project and developed the DES fractionation, lignin and cellulose separation and lignin conversion. Also Y.L. performed experiments related to all stages of the work, collected and analyzed data and wrote the first draft. H.Y. contributed to figure corrections and SEM analysis. N.D. performed in-depth NMR and spectroscopic studies regarding the nature of the DES and contributed to writing the manuscript and to figures and tables, Z.W. pretreated the birch lignocellulose and isolated the milled wood lignin. Z.W. and Y.L. performed the composition analysis and enzymatic hydrolysis experiments. L.H. and E.J. contributed to the hemicellulose analysis. P.J.D. contributed to data analysis and the writing of manuscript, preparation of Figures and prepared several lignin model compounds. K.B. contributed to data analysis, conceived and supervised the research, and wrote the manuscript. All authors commented on the manuscript. Y.L. would like to acknowledge Dr. Maxim Galkin for the useful discussions, Dr. Anastasiia Afanasenko and Douwe S. Zijlstra for the help with the NMR spectroscopy. The authors also thank A. Bakker for providing the birch wood used in Europe.

## Competing interests

The authors declare no competing interests.
