## [Peer Review File · Nature Communications]

REVIEWER COMMENTS

The manuscript by Liu, Yu, Wang, Deuss, and Barta reports an interesting and original investigation on tunable and functional DESs for biomass fractionation and lignin depolymerization. The authors explore a ternary DES system that can be readily tuned depending on the targeted properties of the final products. This manuscript presents some new findings (but somehow expected) of some significance to the biorefinery field, and be of interest to a broad scientific audience. In my opinion, this work provides a significant, conceptually novel insight into this field. However, I feel there are several points the authors should correct, revise, and/or update before publication is recommended.

Major points

- The authors claim that the ternary DES used in this work is unprecedented. However, the same DES (ChCl-OA-EG (1:1:3)) has been reported previously. Please see the following publication. (doi.org/10.1080/01496395.2016.1247864). This means that the ternary DES system studied in this work is not novel.
- Also, there are several previous works on the application of ternary DESs (doi.org/10.1016/j.biortech.2019.01.126) (doi.org/10.1016/j.indcrop.2020.112357). Please add in-depth reviews and the significance of the ternary DES system to the introductions.
- Ethylene glycol participates (is consumed) in the stabilization of lignin structure during the fractionation, which can affect the composition and properties of DES. As can be seen in the NMR spectra of DESs (fresh and recycled), the ratio between ChCl and HBDs visually decreases from the recycled DES. An in-depth analysis of the DESs (fresh and recycled) is highly recommended. (e.g., the ratio of HBA and HBDs over recycling time, purity, acidity, etc.).
- Another question arises at this point. Typically, the primary force that allows DES formation is a strong hydrogen bond between HBA and HBD. EG that used as a HBD seems to be readily dissociated from the DES and participates lignin stabilization. Comprehensive insights towards the specific roles of HBA and HBD is necessary.
- Cost of DES calculation: This can be a misleading interpretation because the price for DES synthesis is not counted.
- Regarding the hydrogenolysis of recovered lignin, it looks like the authors only analyzed the monomers. It would be better to compare the thermal degradability of the lignin samples if the authors provide mass balance analysis (e.g., the yield of solid, liquid, and gas).
- Although the authors claimed relatively high hemicellulose retention, these values are still low. An in-depth discussion on hemicellulose is highly required. Since hemicellulose is the weakest biomass component, it is important to analyze hemicellulose behavior during the DES fractionation. It might interact with DES.
- Several previous works reported high glucose yield from ChCl-OA DES pretreated biomass. (doi.org/10.1016/j.biortech.2016.07.026) Comparing with the lower glucose yield (less than 5% in this work), this seems to require discussion.
- In the enzymatic hydrolysis of the CR, CR obtained from the DPCR20 or CSCR showed low glucose yield, although the more lignin was removed compared to the DPCR10. The authors claimed that enzyme accessibility is more affected by surfaced pseudolignin, not

the total amount of lignin, which is questionable. Also, it is not sure if the authors meant just "condensed lignin" or "pseudo lignin." Considering that a significant portion of hemicellulose is missing, it might form pseudo lignin under acidic conditions. Need more analysis and discussion on it.

- In ref 54, they claimed "lignin droplet," not "pseudo lignin." Also, a high magnification SEM image (0.5 um, as can be seen in ref 54) is needed to better visualize this condensed or pseudo lignin.

Other points

- The title only contains "lignin depolymerization." Please consider revising the title more comprehensively, containing valorization of carbohydrates too. (e.g., total utilization)
- In the abstract, authors claim that "undesired lignin condensation is a critical drawback for DES-based fractionation process." This is a clearly false statement. Depending on what kind of DES system used in the biomass fractionation, lignin condensation can be avoided. If an acidic hydrogen bond donor is used for the synthesis of DES, it could lead to such unwanted secondary reactions. However, this cannot be blamed for all DES-assisted biomass fractionation processes.
- In the introduction, the authors claim that "while these systems generally deliver good quality cellulose....". Previously, the work used ChCl-LA reported highly preserved beta-O-4 structure after biomass pretreatment. Please see the following. (doi.org/10.1039/C8GC03064B)
- In Figure 3a, it is not clear how the sum of selectivity of the final products exceeds 100%. It is not clear how the 5a is formed (where/how did the aldehyde come to C-beta? And then EG attached to 5a?)
- In Figure 5, NMR signals (DPL10) corresponding to beta-5 interunit are missing. Why? And How did authors clearly separate beta-O-4 and beta-O-4' units as these signals overlap each other?
- The authors claim that "However, in these procedures lignin and hemicellulose products are converted..." This statement cannot be agreeable without further TEA.
- In Supplementary Table 8, it is not clear how CR yield exceeds 100%.
- Please provide the calibration data for the individual monomeric phenols.

Reviewer #2 (Remarks to the Author):

The authors developed "a ternary DES" with choline chloride, oxalic acid and ethylene glycol, and tested for birch lignin extraction. There is merit in evaluating "ternary DES" for lignocellulose treatment, although all these three components have been used for DES preparations.

I struggled to understand the purpose and outcome of this study. One objective is to produce lignin that is amiable for depolymerization. The highest lignin extraction yield is 76% (DP DES20); the best depolymerization yield is 22-24% from DPL10 (Hydrogenolysis). The best monomers from initial lignin is less 20%. This is less than a few of existing lignin depolymerization methods and more challenging on an economic sense than these lignin depolymerization methods (e.g. based catalyzed depolymerization, reductive depolymerization, etc.), especially considering the extra process steps involved in DES

extraction, separation and recycling...

The DES recovery is 93%. 32g DES was used for treating 2 g birch (~20% lignin). With a yield of 76% lignin and 20% monomers. It takes 2.24g (32*0.07) of DES to trade ~ 0.06 g of monomers (0.4*0.76**0.2). As author quoted a \$DES \$600-800 per ton, it needs to justify how the monomers' value can compensate this process.

The authors also presented a high retention of cellulose and excellent enzymatic hydrolysis activity. There was no statistics value presented to hydrolysis yield and other compositions. In fact, there is virtually no statistics value for any data presented in this manuscript!

The lignin obtained from a ternary DES appear to be condensed significantly (all shown black color as well).

Response to the reviewer's comments

Reviewer #1

The manuscript by Liu, Yu, Wang, Deuss, and Barta reports an interesting and original investigation on tunable and functional DESs for biomass fractionation and lignin depolymerization. The authors explore a ternary DES system that can be readily tuned depending on the targeted properties of the final products. This manuscript presents some new findings (but somehow expected) of some significance to the biorefinery field, and be of interest to a broad scientific audience. In my opinion, this work provides a significant, conceptually novel insight into this field. However, I feel there are several points the authors should correct, revise, and/or update before publication is recommended.

We appreciate the predominantly positive comments of the reviewer as well as their invaluable suggestions, which we have fully addressed. All necessary experiments were performed to answer the excellent questions of the reviewers.

Major points

Question 1. Ref 1: The authors claim that the ternary DES used in this work is unprecedented. However, the same DES (ChCl-OA-EG (1:1:3)) has been reported previously. Please see the following publication. (doi.org/10.1080/01496395.2016.1247864). This means that the ternary DES system studied in this work is not novel. Also, there are several previous works on the application of ternary DESs (doi.org/10.1016/j.biortech.2019.01.126) (doi.org/10.1016/j.indcrop.2020.112357). Please add in-depth reviews and the significance of the ternary DES system to the introductions.

Answer 1: Indeed, we have previously overlooked the references that described this ternary DES composition, reported chiefly for extraction purposes [Ref 30 and Ref 31 in the main text]. We have now re-worked the introduction according to the reviewer's suggestion. Relevant references to papers, which use the ternary composition (and other ternary systems) in question, were added – see list below. Due to character limitations and the need to extensively discuss relevant compositions, we added a section on this into the Supplementary Section 2, with a clear reference to this in the Introduction part of the main text. The relevant papers in question were cited both in main text and Supplementary Information. Keeping in mind limitation on number of references - more references were added to the Supplementary Information.

We would like to point out that the focus on these papers largely differs from ours, as they utilize ternary DES as solvents for extraction of flavonoids, aqueous ternary DESs were prepared to optimize the extraction efficiency [Ref 30]. Other ternary DES compositions were used for fractionation purposes as specified below. The focus of

our work is very different, and represents an approach which ranges from a fundamental molecular to an engineering/process level.

In our case, the core of the novelty (which, in our view is significant) lies in furthering the concept of ‘task specific alternative reaction media by:

- The incorporation of the function of ‘stabilization’ into the DES.
- Demonstrating tunability of the reaction outcome depending on the DES composition.

More specifically, we demonstrate on lignin models the unique role of EG in our ternary system acting both as DES as well as reagent for either trapping the formed C2-aldehyde species obtained upon β -O-4 acidolysis in the form of EG acetal or capturing the benzylic carbonium ion formed upon dehydration of the β -O-4 motif, while achieving tunability of the outcome of these reactions depending on the composition of the respective DES (CS DES *versus* DP DES). In addition we show that the protection concept can be extended to the use of real lignocellulose feeds, facilitating fractionation, achieving lignin and cellulose stabilization and suppressing lignin redepositing. We also show efficient depolymerization of the obtained lignin and cellulose feeds and a novel method for DES recycling (in high yield and purity). In addition, after revisions, a deeper insight into the nature of the interactions between the components of the ternary DES system (versus a binary DES) is provided by DFT calculation and molecular simulation as well as experimentally by special NMR experiments (NOESY, DOESY) and derived calculations.

Indeed, the DES using the same ternary compositions that have been added to the list of references in the manuscript, are as follows:

- **Ref 30 (main text):** Tang, W., et. al. Evaluating ternary deep eutectic solvents as novel media for extraction of flavonoids from *Ginkgo biloba*. *Separation Science and Technology*, **2017**, 52(1), 91-99.

This paper deals with the extraction of flavonoids using aqueous ternary DES (ChCl: OA: EG).

- **Ref 31 (main text):** Jiang, Z. et. al. Green and efficient extraction of different types of bioactive alkaloids using deep eutectic solvents. *Microchemical Journal*, **2019**, 145, 345–353.

This paper screened a total of 75 types of binary or ternary DESs (including one ternary combination with ChCl: OA: EG) for morphinane alkaloids extraction.

These two papers specifically mentioned by Rev 1 deal with *other types of ternary compositions*:

- **Ref 21 (main text):** Chen Z et. al, Ternary deep eutectic solvents for effective biomass deconstruction at high solids and low enzyme loadings. *Bioresource Technology*, **2019**, 279, 281-286.

Here, the ternary DES (guanidine hydrochloride/EG/p-TSA and also ChCl/EG/p-TSA) **different from ours**, was used for lignocellulose fractionation showing xylan and lignin removal from switchgrass and cellulose retention. The 2D NMR of the lignins showed significant change in lignin interunit linkages.

- **Ref 12 (Supplementary Section 2.1):** Ji Q., et. al. Efficient removal of lignin from vegetable wastes by ultrasonic and microwave-assisted treatment with ternary deep eutectic solvent. *Industrial Crops and Products*, **2020**, 149, 112357.

In this paper, the ternary DES (ChCl-Glycerol- $\text{AlCl}_3 \cdot 6\text{H}_2\text{O}$) **different from ours**, was used for removing lignin from garlic skin and green onion root. No lignin characterization or tunability of the system was shown.

Other recent ternary or relevant DES literature:

The following 2 papers have been cited in the supporting information:

- **Ref 13 (Supplementary Section 2.1):** Xue B. et al. Efficient dissolution of lignin in novel ternary deep eutectic solvents and its application in polyurethane. *International Journal of Biological Macromolecules*, **2020**, 164, 480–488.

In this paper, Ternary DES (ChCl:Gly:PEG-400) was used to dissolve lignin for polyurethanes application, the dissolution of lignin in DES didn't change the lignin structure.

- **Ref 14 (Supplementary Section 2.1):** Chen Z. et al. Insights into Structural Changes of Lignin Toward Tailored Properties during Deep Eutectic Solvent Pretreatment. *ACS Sustainable Chem. Eng.* **2020**, 8, 9783–9793.

In this paper, they used H_2SO_4 acidified ChCl/EG DES for switchgrass fractionation to obtain lignin, the lignin characterized by 2D NMR showed preserved $\beta\text{-O-4}$ linkages (highest 40.3%).

The following papers are 3 component DES, but not strictly relevant and due to restrictions in Nr. of references have been cited in the Supplementary Information.

- **Ref 15 (Supplementary Section 2.1):** Jiang J. et al. High Production Yield and More Thermally Stable Lignin-Containing Cellulose Nanocrystals Isolated Using a Ternary Acidic Deep Eutectic Solvent. *ACS Sustainable Chem. Eng.* **2020**, 8, 7182–7191.

This paper described the use of ternary DES (ChCl: OA: p-TSA) for lignin containing cellulose nanocrystals production.

- **Ref 16 (Supplementary Section 2.1):** Saputra R. et al. Synthesis and thermophysical properties of ethylammonium chloride-glycerol-ZnCl₂ ternary deep eutectic solvent. *Journal of Molecular Liquids*, **2020**, 310, 113232.

This paper mainly focuses on synthesis and thermophysical properties of ternary DES (ethylammonium chloride/glycerol/ZnCl₂).

- **Ref 17 (Supplementary Section 2.1):** Farajzadeh M. A. et al. Preparation of a new three-component deep eutectic solvent and its use as an extraction solvent in dispersive liquid-liquid microextraction of pesticides in green tea and herbal distillates. *J Sci Food Agric* **2020**; 100, 1904–1912.

This paper described the use of ternary DES (dichloroacetic acid, L-menthol, and n-butanol) for extraction of pesticide residues from green tea.

- **Ref 18 (Supplementary Section 2.1):** Fu N. et al. Ternary choline chloride/caffeic acid/ethylene glycol deep eutectic solvent as both a monomer and template in a molecularly imprinted polymer. *Journal of Separation Science*, **2017**, 40, 2286-2291.

This paper described the use of ternary choline chloride/caffeic acid/ethylene glycol deep eutectic solvent as both a monomer and template in synthesizing a molecularly imprinted polymer.

Question 2, Ref 1: Ethylene glycol participates (is consumed) in the stabilization of lignin structure during the fractionation, which can affect the composition and properties of DES. As can be seen in the NMR spectra of DESs (fresh and recycled), the ratio between ChCl and HBDs visually decreases from the recycled DES. An in-depth analysis of the DESs (fresh and recycled) is highly recommended. (e.g., the ratio of HBA and HBDs over recycling time, purity, acidity, etc.).

Answer 2 (Ref 1): The authors thank the Reviewer #1 for the excellent question. We agree, since EG participates in the reaction, it is necessary to quantify these losses. We have now performed several new measurements.

- ¹H NMR analysis of (**DP10 DES**) before and after recycling (**RDP10 DES** = recycled DES), using guaiacol as external standard to account for losses of **EG** and **ChCl**.
- Tracking the pH values prior and post–recycling to evaluate the loss of **Oxalic Acid**
- Estimated purity determination based on additional semi-quantitative ¹H NMR experiments
- Evaluating the performance of the recycled **RDP10 DES** in a next round of fractionation, by determining lignin yield and 2D NMR of the obtained lignin

The following statement/explanation was added to the manuscript: ‘Naturally, during EG protection, a portion of the EG present in the DES is incorporated into

the lignin structure, thus gets consumed during fractionation, therefore the change of DES composition upon fractionation and recycling was carefully assessed (Supplementary Section 7). Overall, these experiments revealed, that the loss of **ChCl** is minimal, while the loss of **EG** and **OA** are more significant, but quantifiable after the first round, and can be compensated by simply adjusting the composition of **RDP10 DES** by adding the appropriate amounts of these components. Moreover, the purity of the recycled DES remains high as stated before. The **RDP10 DES** even with lower **EG** and **OA** content can be reused with a good fractionation efficiency.'

The corresponding experiments have been added to Supplementary Section 7 and Supplementary Figure 50-54, Supplementary Table 19-20.

Question 3, Ref 1: Another question arises at this point. Typically, the primary force that allows DES formation is a strong hydrogen bond between HBA and HBD. EG that used as a HBD seems to be readily dissociated from the DES and participates lignin stabilization. Comprehensive insights towards the specific roles of HBA and HBD is necessary.

Answer 3 (Ref 1): The authors appreciate this excellent comment related to the roles of HBA and HBD and more specifically, the role of EG is the ternary DES. First, we would like to point out that in our view, the involvement of a DES component in a hydrogen bonding interaction should not exclude the possibility for its participation in chemical reactions, where strong covalent bonds are formed. In our tunable ternary DES, EG participates in the formation of either a *cyclic acetal* or *benzyl ether* depending on composition/conditions.

Nonetheless, we agree that the nature of the ternary solvent system should be clarified better, especially given the scarce literature data (see also Answer 1) and very little characterization available thus far. We performed special experiments as well as computational studies to provide better fundamental understanding.

NMR experiments: carried out the characterization of 3 different DES. The known binary systems ChCl-OA and ChCl-EG were compared to the ternary composition in question OA-ChCl-EG. Characterization of the ¹H NMR spectra of the neat DES have shown that the protons of –OH (both EG and ChCl) and –COOH (in OA) were coalescent under room temperature and NOESY spectra revealed a range of correlations that correspond to inter- or intramolecular interactions. Of particular interest was a cross-peak showing correlation between –CH₃ in ChCl and –CH₂- in EG, clearly indicating close proximity of these two components in the (ChCl:EG:OA) ternary DES. Next, to gather more proof, DOSY experiments were performed with all 3 different compositions. Based on these experiments it was concluded that the aggregates in the ternary system were larger, pointing to a 3-component interaction

while in the binary mixtures 2 component aggregates were seen (Supplementary Section 5.1.1). In addition, it was also revealed that the interactions were strongest between ChCl – OA and weaker in the case of ChCl – EG. In addition, DFT calculations and molecular simulations were performed, the interaction energy of ChCl/OA and ChCl/EG are -25.63 KJ/mol and -5.67 KJ/mol, respectively. The results confirm a stronger interaction between ChCl-OA and weaker in the case of ChCl-EG. A comment related to the nature of the system and EG participation in stabilization has been added to the main text. An entire section (See Supplementary Section 5, Figures 19-27) was added for more insight details.

Question 4, Ref 1: Cost of DES calculation: This can be a misleading interpretation because the price for DES synthesis is not counted.

Answer 4, Ref 1: We thank the referee for the excellent comment. We have followed the existing literature in this regard. In the paper of Słupek E. et. al. in Energies, 2020, 13, 3379 a cost calculation for ChCl/OA and ChCl/urea DESs was provided from Alibaba. In the paper of Singh in Green Chem, 2017, 19 (13), 3152-3163 a cost calculation of choline based ionic liquids, was also calculated based on prizes from Alibaba. In the publication of Wan and coworkers in Biofuels, Bioprod. Bioref. 2020, 14, 326–343, the price of **ChCl** was indicated as 1200\$ ton⁻¹ and that of **EG** as 838\$ ton⁻¹ based on Intratec Solutions, LLC, consistent with our values added.

If we understand the question right, the referee would like to have more discussion regarding the price of the DES and we agree this is necessary. The question is both the actual preparation of the DES from its individual components as well as the synthetic effort to arrive to these.

Regarding the procedure to make the DES, it is a very simple practice that can be performed at room temperature by simply mixing of components (see Supplementary Figure 3). We generally selected 80°C to accelerate DES synthesis.

Regarding the cost of the individual components, this depends, and will in the future depend on many factors such as reactants price (and their availability from fossil *versus* renewable resources), synthetic method, scale, and market demands. Therefore, rather than indicating exact prize values, we have now displayed possible synthetic routes for choline chloride and EG from renewable resources, as well as petrochemicals (see Supplementary Figures 1 and 2).

We estimated the prize of the ternary DES in the introduction and a reference to the Supplementary Information leading to the discussion about DES preparation routes was added.

Question 5, Ref 1: Regarding the hydrogenolysis of recovered lignin, it looks like the authors only analyzed the monomers. It would be better to compare the thermal degradability of the lignin samples if the authors provide mass balance analysis (e.g., the yield of solid, liquid, and gas).

Answer 5, Ref 1: Here we have followed the best practices for the lignin depolymerization field (as also recently discussed in Energy Environ. Sci., 2021, 14, 262). In fact, to determine the feasibility of the obtained lignins for aromatic monomers production (which directly depends on the β -O-4 content), the best measure is the proper quantification of the monomers. Due to sufficient preservation of the β -O-4 linkages in our lignin DPL10, the aromatic monomer yields upon depolymerization are high and the reactions display good product selectivity also regarding the types of products we expect from the respective methods. The other side product streams are minimal. The methods used here are widely established and frequently used in our laboratory where these issues were previously quantified [J. Am. Chem. Soc. 2015, 137, 23, 7456–7467].

For reductive depolymerization practices, apart from the monomers, the quantification of the solid residues and gaseous products have been previously evaluated in our laboratories. These generally show that the gas phase consists of >95% H₂ gas (and under these conditions gaseous products that would otherwise be formed by oxidative or thermal processes are not prevalent; and the solids are a small amount of lignin residue mixed with the heterogeneous catalyst for which thermal degradation studies are not suitable).

However, we agree that thermal stability data of the obtained lignins, especially comparisons between the ‘condensed’ and ‘high β -O-4 lignins’ are very useful.

We have now determined the thermal stability two lignin samples (DPL10 and ChCl/OAL) by TG-FTIR, see Supplementary Section 6.3.1, Figures 41 and 42. The yield of solid residues for DPL10 and ChCl/OAL upon degradation at 800 °C was 24.5% and 40.8% respectively, consistent with the degree of condensation. The results of TG-FTIR at 220°C also showed differences: the characteristic bands at 2271-2391 cm⁻¹ and 586-726 cm⁻¹ indicative of CO₂ formation were more pronounced for DPL10 and less intense for ChCl/OAL, as expected from the respective β -O-4 contents and level of condensation.

In addition, lignins ChCl/OAL and DPL10 were further characterized by pyrolysis GC-MS. The stabilized DPL10 lignin, extracted by our ternary DES procedure, clearly demonstrated higher yield of pyrolytic monomers compared to ChCl/OAL, consistent with TG-FTIR analysis. The corresponding data were added into

Supplementary Section 6.3.2, Page 55-56 and Figure 43 and a respective comment was placed to the manuscript.

Question 6, Ref 1: Although the authors claimed relatively high hemicellulose retention, these values are still low. An in-depth discussion on hemicellulose is highly required. Since hemicellulose is the weakest biomass component, it is important to analyze hemicellulose behavior during the DES fractionation. It might interact with DES.

Answer 6, Ref 1: We thank the reviewers for the excellent comment, indeed our main focus has been to establish the concept of stabilization and we focused primarily on quantification of lignin => aromatics and cellulose=> glucose yields and measure of the quality of these streams after fractionation in control DES and our ternary system.

We now have performed a range of additional experiments shown in the supplementary information to discuss the hemicellulose fraction and an in depth discussion was added to the main text as well.

First we performed simple experiments by treating microcrystalline cellulose (MCC) and xylan (as models) separately in different DES. Gratifyingly, MCC was very stable in DP10 DES (99% retention) and displayed high stability in DP20 DES (containing more OA) as well (93.5% retention). This is in good agreement with the lignocellulose fractionation data. When xylan was treated in DP10 DES at 100°C for 24h as hemicellulose model (Supplementary Figure 48c), a clear liquid was obtained after the indicated time, without any coloring, indicating full dissolution of xylan. The quantity of xylose was determined ($7.2\pm 0.42\%$), indicating partial xylose hydrolysis.

Furthermore, in depth sugar composition analysis of the raw birch lignocellulose as well as the cellulose residues obtained upon the various treatments was performed. These showed that hemicellulose suffers debranching upon treatment with all DES systems (loss of Gal, Ara, Rha, GlcA and partial depolymerization to xylose), but remains most intact with DP10 DES. **Hence, our comment regarding high hemicellulose retention is confirmed.** It is indeed this DES system (that also preserves the β -O-4 content of lignin) which leads to the highest retention of sugars. With increasing OA content and especially without EG stabilization (such as in the case of ChCl/OA) not only the lignin is heavily modified but also sugar loss is much more significant.

A discussion was added into the main text and the corresponding experiments have been added to Supplementary Section 6.6, Figure 48 and Table 17.

Question 7, Ref 1: Several previous works reported high glucose yield from ChCl-OA DES pretreated biomass. (doi.org/10.1016/j.biortech.2016.07.026) Comparing with the lower glucose yield (less than 5% in this work), this seems to require discussion.

Answer 7, Ref 1: We thank the reviewer for this nice comment, in our latest results, the highest glucose yield from ChCl/OA CR is ~58%. Indeed the reported values differ from system to system, two parameters being crucially important: the lignocellulose source and the processing conditions. The processing in the mentioned paper is at 90 °C for 24h and the substrate is corncob while we use birch lignocellulose and 120 °C for 0.5h. A comment to the main text regarding the other system has been added. In addition, we are confident in the glucose yield values obtained, as the data have been re-assessed for reproducibility (see also Answer 3, Ref 2)

Question 8, Ref 1: In the enzymatic hydrolysis of the CR, CR obtained from the **DPCR20** or **CSCR** showed low glucose yield, although the more lignin was removed compared to the **DPCR10**. The authors claimed that enzyme accessibility is more affected by surfaced pseudolignin, not the total amount of lignin, which is questionable.

Answer 8, Ref 1: We appreciate this excellent comment of the reviewer, and agree that this has to be elaborated on much more extensively. We broke down the question in two key factors that may affect enzymatic hydrolysis: **1.)** the presence of residual lignin (connected to the degree of delignification) and **2.)** possible recondensation phenomena, that may manifest by objects also seen in SEM, which may be of (hemi)cellulose and of lignin origin (pseudo lignin or condensed lignin).

Comment related to point 1.) We agree that the lignin content may have an effect, but there are several studies that demonstrated high glucose yield (~90%) with 50-60% delignification (Ref 17, 53 and 54 in the main text), even (91% glucose yield) at~10% delignification (Ref 55 in the main text).

- **Ref 54:** Procentese, A. et al. Deep eutectic solvent pretreatment and subsequent saccharification of corncob. *Bioresource Technology*, **2015**, 192, 31-36. [ChCl-imidazole DES, ~50% delignification, 85% glucose yield]
- **Ref 53:** Satlewal, A. et al. Assessing the Facile Pretreatments of Bagasse for Efficient Enzymatic Conversion and Their Impacts on Structural and Chemical Properties. *ACS Sustain. Chem. Eng.* **2019**, 7, 1095–1104. [ChCl-Lactic acid DES, ~50% delignification, 86% glucose yield]
- **Ref 17:** Guo, Z. et al. Short-time deep eutectic solvent pretreatment for enhanced enzymatic saccharification and lignin valorization. *Green Chem.*

2019, 21, 3099–3108. [BTMAC-Lactic acid DES, ~63% delignification, 91% glucose yield]

- **Ref 55:** Ai, B. et. al. Natural deep eutectic solvent mediated extrusion for continuous high-solid pretreatment of lignocellulosic biomass *Green Chem.*, 2020, 22, 6372-6383. [ChCl/Glycerol DES, ~10% delignification, ~90% glucose yield]

Based on this literature precedence, we do not connect the delignification values with the glucose yields and are confident with the yields, as reproducibility of these experiments was further assessed. See also Supplementary Tables 12-14 and 17.

According to these new data, the delignification of DPCR10 and DPCR20 is 54.3 ± 3.0 % and 70.4 ± 5.5 %, respectively. The glucose yield values are high in both cases: DPCR20 (92.5 ± 1.63 %) and DPCR10 (95.9 ± 2.12 %). The minor difference may be due to the more condensed lignin in the CR matrix of DPCR20.

Comment related to point 2.) We repeated the fractionation and enzymatic hydrolysis experiments. Nanosized objects were again confirmed on ChCl/OA CR (control without stabilization) while these objects were not seen in DPCR10. Delignification for both these CR was largely similar (~50%), while the glucose yield of DPCR10 was significantly higher than ChCl/OA CR. This strongly supports the notion that these objects (most likely condensed lignin, see below) are responsible for lower enzymatic hydrolysis activity. More details regarding the nature of these objects is found in Answer 9 below. A discussion, alongside the references above, has now been added to the main manuscript text.

Question 9, Ref 1.: Also, it is not sure if the authors meant just "condensed lignin" or "pseudo lignin." Considering that a significant portion of hemicellulose is missing, it might form pseudo lignin under acidic conditions. Need more analysis and discussion on it.

Answer 9, Ref 1. We thank Reviewer 1. for the insightful comment and agree that this point needs more clarification/investigation. We have now performed a series of control experiments (Supplementary Section 6.4, Figures 45 and 46) using microcrystalline cellulose (MCC) and several model compounds. In these experiments, MCC alone or MCC in the presence of xylan (as hemicellulose model) or lignin was treated under the reaction conditions resembling lignocellulose fractionation in either ChCl/OA DES (120°C for 30 min) or DP10 DES (100°C for 24 h). When MCC or a combination of MCC and xylan (0.45g MCC and 0.35g xylan) were fractionated in ChCl/OA DES, no nanosized particles were observed by SEM on the recovered cellulose residues (white color). However, when the MCC+xylan+lignin (0.45g MCC, 0.35g xylan and 0.2g lignin) combination was treated in ChCl/OA DES, obvious nanosized particles were formed on the surface of the residues (dark brown). When

same experiments were performed in DP10 DES, the cellulose residue (light brown and fine surface) showed a relatively “clean” surface without any particles, the cellulose appeared fibrillated into nanofibers. These results were also consistent with SEM analysis of the cellulose residues from the corresponding lignocellulose fractionation experiments. These experiments strongly suggest that the nanosized particles are condensed lignin. The ability of DP10 DES to ‘protect’ the lignin from condensation is likely responsible for this behavior, compared to ChCl/OA DES with higher acid content.

We have now changed our original comments referring to pseudolignin to condensed lignin and made further textual changes. Also a short discussion into the manuscript and the Supporting information regarding the extra experiments was added. We also changed the position of the references.

Supplementary Figure 46. SEM images of MCC solid residues obtained with different treatments and additives. **Upper row:** low magnification, **Bottom row:** high magnification. **(a and b):** MCC (0.45g MCC) treatment in ChCl/OA DES; **(c and d):** MCC and xylan (0.45g MCC and 0.35g xylan) treatment in ChCl/OA DES; **(e and f):** MCC+xylan+lignin (0.45g MCC, 0.35g xylan and 0.2g lignin) treatment in ChCl/OA DES; **(g and h):** MCC+xylan+lignin (0.45g MCC, 0.35g xylan and 0.2g lignin) treatment in DP10 DES; **Inset:** digital pictures of the solid residues; the recovery of the solid residues and treatment conditions (120°C, 30min for ChCl/OA DES and 100°C, 24h for DP10 DES) follow the same procedure as the lignocellulose fractionation process.

Question 10, Ref 1.: In ref 54, they claimed "lignin droplet", not "pseudo lignin." Also, a high magnification SEM image (0.5 μm , as can be seen in ref 54) is needed to better visualize this condensed or pseudo lignin.

Answer 10, Ref 1.: We thank the reviewer for the excellent comment, and we agree that there needs to be a better clarification of the issue of pseudolignin versus condensed lignin. The ChCl/OA CR and DPCR10 were further characterized by SEM at high magnification (Figure 7 in the main text). It has been confirmed that both the high resolution as well as the low resolution images look different for the control ChCl/OA versus DP10 fractionated cellulose residues. The sample obtained upon DP10 fractionation shows a fibrous cellulose structure with no significant amount of nanosized particles. Based on the control experiments in Answer 9 we believe that these objects are condensed lignin.

Regarding Ref 54, we are sorry for the mistake. Reference 54 is a beautiful work containing high quality imaging of lignin droplets, and focusing on the morphology of these droplets, hence it was cited in connection with SEM work. We also would like to point out that our nanosized objects are generally in a smaller size range and with slightly different morphology compared to the ones observed in ref 54, this is understandable given the different processing conditions used.

Part of Figure 7. Representative SEM images of the DES fractionated birch cellulose residues. SEM images of ChCl/OA DES fractionated CR at low magnification (c) and high magnification (d), showing obvious lignin particles formation (condensed lignin which is confirmed as shown in Supplementary Section 6.4.1, average diameter of ~ 54 nm) on the surface; DP10 DES fractionated CR at low magnification (e) and high magnification (f), the cell structure of DPCR10 was fibrillated and without formation of lignin particles.

Other points

Question 11, Ref 1.: The title only contains "lignin depolymerization." Please consider revising the title more comprehensively, containing valorization of carbohydrates too. (e.g., total utilization)

Answer 11, Ref 1.: We agree, the title has been revised to reflect a more general lignocellulose valorization. We would like to refrain from saying ‘ full’ or ‘ total’ as this would be an ideal system.

Question 12, Ref 1.: In the abstract, authors claim that "undesired lignin condensation is a critical drawback for DES-based fractionation process." This is a clearly false statement. Depending on what kind of DES system used in the biomass fractionation, lignin condensation can be avoided. If an acidic hydrogen bond donor is used for the synthesis of DES, it could lead to such unwanted secondary reactions. However, this cannot be blamed for all DES assisted biomass fractionation processes.

Answer 12, Ref 1.: The abstract and introduction has been revised accordingly. Also Figure 1 has been corrected. We now made it clear that we refer to acid based systems.

Question 13, Ref 1.: In the introduction, the authors claim that "while these systems generally deliver good quality cellulose....". Previously, the work used ChCl-LA reported highly preserved beta-O-4 structure after biomass pretreatment. Please see the following. (doi.org/10.1039/C8GC03064B)

Answer 13, Ref 1.: We thank the referee for this valuable comment, and agree that this point needs more clarification. For the reference in question, it was claimed that the β -O-4 structure of the lignin was preserved when using ChCl-LA DES at milder conditions, we cannot, however fully agree with this statement. Also in this system, there is significant lignin condensation, as it is apparent from the Figure 6 and Table S1 provided in this publication. The yield of lignin was still relatively low at milder conditions: 6.4% (at 90°C) and 18.5% (at 100°C) and the content of β -O-4 is not calculated from 2D NMR in the paper. When the temperature increased to 130 °C, the β -O-4 content drastically decreased (Figure 6).

In our view – this lignin behavior is completely expected – and it is no surprise that most DES fractionation works focus on the cellulose valorization or bulk lignin valorization but no lignin depolymerization, and that to the very best of our knowledge, in the vast majority of papers there is no full/or proper lignin characterization by 2D HSQC NMR. **As it has been put forward in a number of recent articles, stabilization pre- or post-depolymerization is a fundamentally**

important concept in the lignin valorization field [see Nature Reviews Chemistry, 2020, 311–330; Energy Environ. Sci., 2021, 14, 262]. Hence we believe the fundamental study of the concept of ‘stabilization’ of reactive intermediates (and of the benzylic carbocation that easily forms even under mildly acidic conditions) and the incorporation of this concept into the field of DES, as demonstrated in this paper, is an important step forward.

Question 14, Ref 1.: In Figure 3a, it is not clear how the sum of selectivity of the final products exceeds 100%. It is not clear how the 5a is formed (where/how did the aldehyde come to C-beta? And then EG attached to 5a?)

Answer 14, Ref 1.: The reviewer makes an excellent point, in that the mechanism of C2-aldehyde and acetal formation is not trivial to see. We have extensively discussed the C2- and C3-pathway in our previous works, which have been readily cited. A mechanistic scheme is now provided in the Supplementary Figure 4. On Figure 3a we also show that 5a is formed by rehydration of the enol ether intermediate, the cleaved products are aldehyde (5a) and guaiacol followed by trapping of unstable 5a in the form of cyclic acetal 6aa by the ethylene glycol in DES.

Regarding the products, in the **ideal case**, cleavage of the intermediate 3a leads to the clean formation of 2 products, namely acetal (6aa) and guaiacol in the presence of ethylene glycol stabilization (with both 100 % yield and 100% selectivity each. In the absence of stabilization 2 main products, namely 5a and guaiacol are formed, these however don't tend to reach high selectivity due to the formation of intractable humins.

Question 15, Ref 1.: In Figure 5, NMR signals (DPL10) corresponding to beta-5 interunit are missing. Why? And How did authors clearly separate beta-O-4 and beta-O-4' units as these signals overlap each other?

Answer 15, Ref 1.: Both related figures in the manuscript and the supporting information have now been corrected. The β -5 linkages can now be observed, albeit as weak signals. This is due to the relatively low β -5 content of birch lignin.

Regarding the signals of the β -O-4 γ and β' -O-4 γ – we have re-labelled these signals as overlapping β -O-4 γ / β' -O-4 γ (Figure 6b), other 2D HSQC NMR signals of β -O-4 and β' -O-4 units can be distinguished when d_6 -acetone is used for measurement instead of commonly used d_6 -dmsO as described in [Green Chem., 2015, 17, 4980].

Also, there was a numbering mistake, Figure number was updated from Figure 5 to Figure 6.

Figure 6 Comparison ^1H - ^{13}C HSQC NMR spectrum of mill wood lignin (MWL) and DPL10 from birch lignocellulose. ^1H - ^{13}C cross signals in the side-chain region of MWL (a) and DPL10 (b), showing obvious ethylene glycol incorporation signals of β' -O-4 which means ethylene glycol stabilize the benzylic carbocation of β -O-4 (the ethylene glycol incorporation signals were confirmed by overlapping the HSQC NMR spectrum of lignin and model compound) and aromatic region of MWL (c) and DPL10 (d) with typical S and G signals, slight condensation (14-16%) signal was observed in the aromatic region of DPL10. Spectra of both MWL and DPL10 were recorded by dissolving the corresponding lignin in d6-acetone for measurement instead of commonly used d6-dms0 (several drops of D_2O if necessary).

Question 16, Ref 1.: The authors claim that "However, in these procedures lignin and hemicellulose products are converted..." This statement cannot be agreeable without further TEA.

Answer 16, Ref 1.: We agree with the reviewer – without proper TEA it is better not to compare the two methods. We have now rephrased the sentence, it was not necessary to add this comment.

Question 17, Ref 1.: In Supplementary Table 8, it is not clear how CR yield exceeds 100%.

Answer 17, Ref 1.: Yes, it was found that the CR yield exceeds 100%, especially at high EG content. In our recent work on lignin-first fractionation of softwood lignin by acidolysis/EG stabilization [ChemSusChem, 2020, 13, 4468-4477.], we have also observed incorporation of EG (which is in excess) into various sugar fractions (monosaccharides, oligosaccharides and lignin oligomers). It is likely that EG is either

adsorbed or chemically bound to CR via Fisher-glycosylation in the C1 position (as confirmed by model studies in our previous work mentioned above). A footnote for explanation alongside with this reference was added to the Supplementary Table 12.

Question 18, Ref 1.: Please provide the calibration data for the individual monomeric phenols.

Answer 18, Ref 1.: The calibration data for the monophenols have now been added to the Supplementary Figure 49.

Reviewer #2 (Remarks to the Author):

The authors developed “a ternary DES” with choline chloride, oxalic acid and ethylene glycol, and tested for birch lignin extraction. There is merit in evaluating “ternary DES” for lignocellulose treatment, although all these three components have been used for DES preparations.

Question 1, Ref 2.: I struggled to understand the purpose and outcome of this study. One objective is to produce lignin that is amiable for depolymerization. The highest lignin extraction yield is 76% (DP DES20); the best depolymerization yield is 22-24% from DPL10 (Hydrogenolysis). The best monomers from initial lignin is less 20%. This is less than a few of existing lignin depolymerization methods and more challenging on an economic sense than these lignin depolymerization methods (e.g. based catalyzed depolymerization, reductive depolymerization, etc.), especially considering the extra process steps involved in DES extraction, separation and recycling

Answer 1, Ref 2.: We thank Reviewer 2 for their valuable comments. The main purpose of this study is to pave the way toward intelligent, tailor made, alternative reaction media, by judiciously incorporating the ‘stabilization’ function that has emerged as one of the prime concept for lignin valorization - into the composition of the DES. In our opinion, the fact that the reactivity and product outcome can be tailored toward either β -O-4 linkage scission followed by C2-aldehyde trapping as acetal *versus* benzylic carbocation protection via the formation of an ether linkage depending on the composition of this DES is one of the most important outcomes of this study. We believe that this is a significant advance since most studies focus on trial and error mixing of DES components and their use in bulk processes such as extraction of exotic plants/compounds, or lignocellulose fractionation without providing little structural insight or mechanistic rationale, especially related to lignin.

Furthermore, we have identified that protection of the benzylic carbocation also works in conjunction with lignocellulose fractionation, leading to widespread preservation of the β -O-4 linkage, supported by NMR studies, and leading to the isolation of ‘protected lignins’ which can be used for materials or depolymerization purposes, or making materials in the future.

It is also important that the stabilization due to the use of DP10 DES allowed to preserve the cellulose quality compared to condensed lignin in the control reaction, this leads to excellent, up to >95% glucose yields and a high hemicellulose retention is also seen.

Lignin yield can be still improved by future optimization, but the β -O-4 content of the obtained lignin is excellent, represents stabilization in its near-native form. As consequence, the depolymerization results obtained in this paper, are excellent, not far behind of the best systems, e.g. those obtained by RCF (here H₂ gas, metal catalyst, 200-220°C are being used).

It is of no surprise, that DES isolated lignins have not been frequently considered /studied for depolymerization purposes. Naturally, many of these systems contain acid to enhance fractionation, which automatically leads to lignin condensation. The issue of condensation has been recently also summarized in our perspectives in [Nature Reviews Chemistry, 2020, 311–330; Energy Environ. Sci., 2021, 14, 262], now cited. Because condensation is unavoidable without stabilization, this per definition decreases the useful monomer yield. To the best of our knowledge, these are the highest aromatic monomer yields obtained from DES isolated lignin.

In addition, our paper provides a range of novelties, including a new DES isolation/recycling process that saves solvent and water demand significantly compared to most recycling approaches.

To address the reviewer’s point better, we have now added a discussion to the main text and a comparison in the Supplementary Information, alongside with a number of references/works/conditions.

Question 2, Ref 2.: The DES recovery is 93%. 32g DES was used for treating 2 g birch (~20% lignin). With a yield of 76% lignin and 20% monomers. It takes 2.24g (32*0.07) or DES to trade ~ 0.06 g of monomers (0.4*0.76**0.2). As author quoted a \$DES \$600-800 per ton, it needs to justify how the monomers’ value can compensate this process.

Answer 2, Ref 2.: We thank Reviewer #2 for the excellent comment. But it is not only DES cost that matters, it is the total LCA and TEA that needs to be performed to see cost invested versus profit from all products from the biorefinery setting. We have

here outperformed, in several aspects, conventional biorefinery approaches. We have excellent overall lignocellulose valorization (both lignin and cellulose streams), by maintaining the value of both cellulose and lignin constituents after processing. This includes the high quality cellulose residues that give excellent glucose yield ($95.9 \pm 2.12\%$, $\sim \text{€}450/\text{ton}$) and high quality lignin. Specifically, from the protected lignin here obtained, we have a 6 times higher monomer yield compared to the lignin obtained in the control reaction. Moreover, the monomers are obtained in very high selectivity which will make any future product separation/purification easier, and more feasible. These aromatic monomers can be converted to any number of attractive products, such as phenol (~ 11 million metric tons in 2019 (IHS Markit's Chemical Economics Handbook-Phenol), $\text{€}1300/\text{ton}$), but higher value products can be targeted as well. Especially, the C2 acetal obtained from DP10L upon acidolysis and protection is a primary platform for the production of dopamine based pharmaceuticals (the cost of dopamine is $\text{\$}29310/\text{Kg}$ (<https://www.drugs.com/price-guide/dopamine>)). Most of these aspects are already present in the depolymerization section, and now a comment was added into the manuscript, conclusion part.

Question 3, Ref 2.: The authors also presented a high retention of cellulose and excellent enzymatic hydrolysis activity. There was no statistics value presented to hydrolysis yield and other compositions. In fact, there is virtually no statistics value for any data presented in this manuscript!

Answer 3, Ref 2.: We agree with reviewer #2 about the need for more extra experiments. We have now performed experiments in triplicate, the results with statistic error bar were updated both in the manuscript and supporting information which confirmed the statements previously made in the paper. The corresponding Figures 6a and 6b (glucose and xylose yield) have been modified in the main text. Moreover, we have also shown that the results show excellent reproducibility in two different laboratories: Northeast Forestry University, China as well as University of Groningen. The Groningen data have been also displayed in the Supplementary Section 6.5.2, Tables 16 and 17.

Question 4, Ref 2.: The lignin obtained from a ternary DES appear to be condensed significantly (all shown black color as well).

Answer 4, Ref 2: The condensation ratio of different lignin was calculated according to the 2D NMR, indeed there's condensation signal in the lignin, but the condensation ratio of the ternary DESs extracted lignin was much less than ChCl/OAL as discussed in the text (**53%** $S_{\text{condensed}}$ of ChCl/OAL *versus* **14%** $S_{\text{condensed}}$ of DPL10). The total β -O-4 content retention for DPL10 was 84%. The lignin obtained from ternary DP10

DES showed highest total aryl ether retention among the DES fractionation systems used in this study and to the best of our knowledge, this is the best overall. The fractionation process was repeated to confirm the reproducibility, similar yield of 44% DPL10 and condensation ratio of only 13.6% was obtained, the lignin showed a high total β -O-4 content of 52.9 per 100 aromatic units (β' -O-4: 44.7 and β -O-4: 8.2) displaying excellent reproducibility. The new result was added to the Supplementary Figure 55.

REVIEWERS' COMMENTS

Reviewer #1 (Remarks to the Author):

The authors addressed all the comments and questions raised by this reviewer. Especially, considering the complex structure of DESs, the authors seem to put extra efforts to elucidate the structure of DESs, using advanced NMR techniques. Overall, this reviewer believes that this work now meets all the rigorous standards of NCOMM and I recommend this to be published.

Here are some additional points for future references.

- For NMR analysis of DESs, the synthesized DESs were mixed with D₂O. Considering typical sample prep of NMR analysis, the amount of solvent is pretty high, which means there will be an interaction between the solvent and DES components. There's a coaxial NMR insert tube the solvent and the samples are separate and can remove any possibility of unexpected interactions between samples and the solvent.

Please note that this is my suggestion for future work (not current work)

- NMR analysis cannot detect chloride ion-associated effect. Although DFT calculation can observe the effect/role of chloride ions on the formation of DESs, the limitation should be noted.

Reviewer #2 (Remarks to the Author):

Authors showed EG when combined with ChCl-OA can protect some beta ether linkages during lignin extraction. But this gave a relatively lower lignin extraction yield (~35%) with partial condensation. The subsequent depolymerization gave low monomers yield. 22-24%. EG has a good lignin extraction ability and has been tested for pulping. The authors need to compare the results from EG alone for lignin extraction and depolymerization of extracted lignin. Why DP DES 20 gave such a high SD value? $49.5 \pm 22.2\%$? why treatment with recycled DES, RDP-DES10, gave a higher lignin yield?

The authors commented on the cost issue. Obviously the study is not aiming at making dopamine. The high DES to substrate ratio as well as solvent loss (e.g 30% EG loss) can be a barrier for commodity products. Besides, there is a significant process cost, especially when liquid-liquid extraction is used. The authors need to provide Technoeconomic analysis to support cost argument.

The authors need to provide phase diagram or eutectic point measurement to show indeed this ChCl-OA-EG is a ternary "DES" system.

The discussion on cellulose hydrolysis is weak. SEM does not provide sufficient explanation.

The authors need to determine crystallinity, surface area/pore volume etc. It is striking to see 70% cellulose can give 118% glucose (SI table 12).

Response to the reviewer's comments:

Reviewer #1:

The authors addressed all the comments and questions raised by this reviewer. Especially, considering the complex structure of DESs, the authors seem to put extra efforts to elucidate the structure of DESs, using advanced NMR techniques. Overall, this reviewer believes that this work now meets all the rigorous standards of NCOMM and I recommend this to be published.

The authors thank for the excellent suggestions and comments which clearly made the manuscript better. We also think that we addressed all the comments carefully. The rest of the points are addressed below.

Here are some additional points for future references.

Comment # 1- For NMR analysis of DESs, the synthesized DESs were mixed with D₂O. Considering typical sample prep of NMR analysis, the amount of solvent is pretty high, which means there will be an interaction between the solvent and DES components. There's a coaxial NMR insert tube the solvent and the samples are separate and can remove any possibility of unexpected interactions between samples and the solvent. Please note that this is my suggestion for future work (not current work)

Thank you for pointing this out and we very much agree with this.

Indeed, being aware of the possibility of different interactions in the presence of water, the NMR experiments realized by us for the elucidation of the DES structure (Supplementary Note 4, e.g. Supplementary Figure 19-21) were performed on the neat DES, and D₂O was placed in a closed capillary tube inside the NMR tube.

To comment on the presence of water, we have observed only one signal for all the mobile protons, which happens in the presence of various amounts of water. (protons belonging to the hydroxyl groups of ChCl and EG (4 – ChCl, 7 – EG) and carboxylic acid of OA (5), as well as H₂O eventually from the oxalic acid hydrate. Furthermore, as explained in the supplementary information we have observed some ester formation between ChCl and EG, which also leads to formation of water in situ (also seen in [D. Polomski, P. Garbacz, K. Czerwinski, M. Chotkowski, Journal of Molecular Liquids 327 (2021) 114820; N. Rodriguez

Rodriguez, A. van den Bruinhorst, L. J. B. M. Kollau, M. C. Kroon, K. Binnemans, ACS Sustainable Chem. Eng. 2019, 7, 11521–11528]).

Interestingly, the DES nanostructure is retained to a remarkably high level of water (ca. 42 wt% H₂O) in the system, [Angew. Chem. Int. Ed. 2017, 56, 9782–9785], after a certain dilution the DES structure collapses.

Comment # 2. NMR analysis cannot detect chloride ion-associated effect. Although DFT calculation can observe the effect/role of chloride ions on the formation of DESs, the limitation should be noted.

This is an excellent point, and we agree. Here in this work, NMR analysis (¹H, ¹³C, COSY, HSQC, HMBC, NOESY and DOSY experiments) were used to elucidate possible correlations between protons and also carbons of components. Calculations were done to assess the possible differences and the strength of the interactions between the different HBD and HBA.

The comment of the reviewer was added into the supplementary information, and labelled in blue.

Reviewer #2 :

Comment # 1- Authors showed EG when combined with ChCl-OA can protect some beta ether linkages during lignin extraction. But this gave a relatively lower lignin extraction yield (~35%) with partial condensation. The subsequent depolymerization gave low monomers yield. 22-24%.

We thank the reviewer for this comment. The fractionation of lignocellulose in DP10 DES was repeated at least 5 times only during revision. In fact, the fractionation was repeated in 2 different labs, by 2 different researchers (in China and Netherlands) and the data are gratifyingly very reproducible. We have now re-evaluated the obtained data where there was 1 outlier. We decided to keep the 4 parallel experiments (44%, 41.3%, 40%, 40%), which accounts for 40.0±3.5%. This has been updated in the manuscript.

Regarding the comment on the lignin yield and monomer yield, the β-O-4 content is very high for our lignin and the monomers yield of 24 wt% achieved is to the best of our

knowledge, the highest among known DES treated lignin depolymerisation efforts and comparable with or better than other organosolv lignin hydrogenolysis efforts (Supplementary Table 15).

In classical organosolv processes used, there is a trade-off between the lignin yield and the β -O-4 content. From our own experiences making an organosolv lignin with high enough β -O-4 content (comparable to the values achieved in this paper), will lead to a lignin yield of around 2-10% by organosolv processing in batch mode.

So in summary, we don't agree with the statement that our lignin yield or monomer yields are low. Proper comparison is provided in Supplementary Table 15 and this point has been discussed in the manuscript.

Comment # 2- EG has a good lignin extraction ability and has been tested for pulping. The authors need to compare the results from EG alone for lignin extraction and depolymerization of extracted lignin.

The authors are grateful for the reviewer's comment. Generally, higher temperature is required for EG pulping to enhance lignin extraction, for example according to *ACS Omega* **2019, 4, 14, 16103–16110**, a maximum of 27 wt % lignin was obtained using aqueous EG with 1wt% sulfuric acid at 180°C, but also leading to lignins with altered structure. From the conclusion section of this paper: "EG-assisted biomass fractionation resulted in 27 wt % lignin recovery, but it also significantly cleaved aryl–ether bonds in lignin during the process. The use of the sulfuric acid catalyst at a high temperature during the fractionation cleaved most aryl–ether linkages in lignin, which are important to preserve for lignin valorization."

In addition, to completely satisfy the reviewer's concern, we performed an extra lignocellulose fractionation experiment with only EG and oxalic acid, following the same reaction conditions as with DP10 DES (100°C, 24h). Under these conditions the lignin yield was **18.9%, much lower than** what we obtained with the DES systems, clearly showing the benefit of the original DES system. Nonetheless, this is still a good result among organosolv processing efforts.

An interesting observation, maybe valuable for future work, is that oxalic acid seems to be better suited for this kind of extraction than sulfuric acid. The extracted lignin was further characterized by 2D HSQC NMR, the lignin also showed high EG incorporated β -O-4 linkage (40 per 100 aromatic units) with a total β -O-4 content of 47 per 100 aromatic units

and less condensation (8%). This experiment was added to the supplementary information (Supplementary Fig. 55) and labelled in blue.

Supplementary Figure 55. 2D HSQC NMR analysis of lignin from EG pulping with OA

Comment # 3- Why DP DES 20 gave such a high SD value? $49.5 \pm 22.2\%$? why treatment with recycled DES, RDP-DES10, gave a higher lignin yield?

We thank the reviewer for the indication, and we agree the provided error value was too high. We apologize for this mistake. Indeed we have earlier performed the fractionation with DP DES 20 at least five times to make sure of reproducibility, but a high lignin yield was obtained only one time which seems to be an outlier. Now we have removed this outlier and re-calculated the lignin yield for DP20 DES, the yield is $53.3 \pm 10.7\%$ and we have revised the value in the manuscript.

Regarding the RDP10 DES, it gave a lignin yield of 41% and the obtained lignin showed high total β -O-4 content (Table 1, Entry 3, Supplementary Fig. 41). This yield is perfectly comparable with fresh DP10 DES fractionation ($40.0 \pm 3.5\%$). Note that the latter value has also been adjusted (see Comment #1) after careful evaluation of the data.

Comment # 4. The authors commented on the cost issue. Obviously the study is not aiming at making dopamine.

We thank the reviewer for the critical comment; however we believe that our previous answer related to the cost issue was more than exhaustive. We provided good examples of potential valuable products for the valorisation of both the lignin and cellulose streams in this work and indeed, dopamine is one of them. We have already described the production of potentially high value biologically active amines from lignocellulose (in a different system): *ACS Cent. Sci.* **2019**, 5, 10, 1707-1716.

Regarding the production of dopamine (ultra-high value of \$29310/Kg): we point out that the C2-acetal monomer we refer to (Table 3), obtained by acidolysis/stabilization first described by our group, does feature the perfect C2-structure for an eventual one-step, one-pot catalytic conversion to dopamine. Further related work is currently in progress, but we would not like to add any more details on this until these studies are concluded.

Comment # 5. The high DES to substrate ratio as well as solvent loss (e.g 30% EG loss) can be a barrier for commodity products. Besides, there is a significant process cost, especially when liquid-liquid extraction is used. The authors need to provide Teconoecconomic analysis to support cost argument.

Regarding the high DES to substrate ratio: this is a parameter that will be optimized in the future, besides, as we earlier agreed, it is not only the DES cost that matters, it is the total LCA and TEA that needs to be performed – and as earlier mentioned, not only commodity but also higher value chemicals may be targeted. While such a comprehensive analysis is clearly outside the scope of this manuscript, we believe (based on literature) that the work presented here will be highly competitive in terms of LCA/TEA, especially since we are also able to preserve the lignin quality. The objective of this work rooted in fundamental scientific understanding, was to provide high quality cellulose and lignin streams that are suitable for efficient downstream processing. This objective has been met, and we believe this represents a significant progress over conventional biorefinery approaches and ‘conventional’ DES acid based fractionations also, which usually lead to more condensed lignins.

Regarding the EG ‘loss’ we cannot call incorporation of EG into the lignin structure a solvent loss. In case the EG protected lignin is used for materials (e.g. for polymers), the EG becomes

part of the product. In case of further downstream processing (e.g. by hydrogenolysis as shown in Table 2), EG will be recycled.

Regarding the liquid-liquid extraction of lignin, our new protocol uses significantly less water (~25 mL water per gram lignocellulose versus classical isolation methods ~49-600 mL water per gram lignocellulose), and allows for more efficient DES recovery. Moreover, the organic solvent can be easily recovered/recycled by distillation. So in essence, yes we believe that this method represents significant progress compared to other DES fractionations where the use of large amounts of water may indeed be a big problem. The unique, increased polarity of the 'EG protected lignins' made this new procedure possible in the first place.

Regarding TEA/LCA: As mentioned above, this is clearly outside the scope of this manuscript, which is already very rich in new results and technical details. A comprehensive TEA will be subject of further work in collaboration.

A comprehensive TEA for a biorefinery using natural DES (ChCl-LA-H₂O) is reported in recent literature [AK Kumar et al. Techno-economic evaluation of a natural deep eutectic solvent-based biorefinery: Exploring different design scenarios. *Biofuels, Bioprod. Bioref.* 2020, 14:746–763.] This work focuses on DES pre-treatment, and cellulosic ethanol production and includes necessary downstream operations for product extractions, solvent recovery etc.

Based on this work, here are the reasons why we think our system will be highly competitive:

1.) It is calculated that a multi-product biorefinery that considers lignin, xylan and other downstream products, becomes economically feasible. In this work it is clearly stated, that “lignin has the greatest impact on revenue” and “multi-product biorefineries provide better sustainability”.

Since in our work the developed EG stabilization allows to preserve the value of both cellulose and lignin streams, and lignin can be further valorised, this will be highly beneficial for TEA, especially when (as mentioned before) higher value products, such as dopamine can be easily targeted.

2.) It has also been identified that solvent recycling and reuse is a key challenge for feasibility. The new DES recycling and lignin isolation protocol described in this paper, uses significantly less water than conventional DES isolation methods, allows for organic solvent

recycling, and recovery of DES in excellent purity, the composition of which can be easily re-adjusted if necessary. These aspects will surely show beneficial in a future TEA analysis.

Comment # 6. The authors need to provide phase diagram or eutectic point measurement to show indeed this ChCl-OA-EG is a ternary “DES” system.

As extensively summarized in Supplementary Section 2: “**Related ternary DES systems in the literature**”, based on literature, the ternary composition ChCl-OA-EG has already been identified as a ternary DES system. The other 2 closely related compositions ChCl/OA DES and ChCl/EG DES are well-known two-component DES systems.

Comment # 7. The discussion on cellulose hydrolysis is weak. SEM does not provide sufficient explanation. The authors need to determine crystallinity, surface area/pore volume etc.

We appreciate the reviewer’s comment, the extra experiments were performed to determine the crystallinity and surface area of the cellulose residues by XRD and BET. The corresponding data have been included into the supplementary information (Supplementary Fig. 56 and Table 21) and marked blue.

The crystallinities of the untreated wood and fractionated cellulose residues (ChCl/OACR and DPCR10) were characterized by X-ray diffraction (XRD). All the samples showed typical cellulose I structure as shown in Supplementary Fig. 56, the crystallinity of untreated wood was 57% according to Segal’s method, while the ChCl/OACR and DPCR10 showed increased crystallinities of 68% and 70%, respectively. The increase in crystallinity is typical for DES fractionated samples, and is attributed to the removal of lignin and some of the hemicellulose (amorphous components) during fractionation.

Supplementary Fig. 56 XRD curves of untreated wood, ChCl/OACR and DPCR10.

The surface area of the cellulose residues were tested by Micromeritics ASAP2020 gas adsorption analyzer. The results are summarized in Supplementary Table 21. After fractionation in DES, ChCl/OACR and DPCR10 showed increasing BET surface area of 1.68 m²/g and 1.89 m²/g, respectively compared to untreated wood, consistently with literature. Both the surface area and volume of pores of ChCl/OACR were slightly lower than DPCR10. This is further hint at the existence of lignin condensation in the cellulose matrix in the case of the control reaction.

Supplementary Table 21 Surface area and pore volumes determined by BET

Samples	BET surface area / m ² /g	BJH volume of pores / cm ³ /g
Untreated wood	0.62	nd ^a
ChCl/OACR	1.68	0.0029
DPCR10	1.89	0.0036

a: Not detectable

Comment # 8. It is striking to see 70% cellulose can give 118% glucose (SI table 12).

We thank the reviewer's indication. The Supplementary Table 12 (now Supplementary Table 16) is composition analysis of the CRs. It was observed that at 80 °C, the yield of CR exceeds 100%, due to EG incorporation as also seen in Ref 41 (ChemSusChem 2020, 13, 4468-4477.) in the main text. We already put the footnote in Supplementary Table 12 (now Supplementary Table 16) to explain this. The glucose yields were calculated based on the retained cellulose in CRs.